# A strategy for Cas13 miniaturization based on the structure and AlphaFold

Feiyu Zhao[1,4], Tao Zhang[1,4], Xiaodi Sun[1], Xiyun Zhang[1], Letong Chen[1], Hejun Wang[1], Jinze Li[1], Peng Fan[1], Liangxue Lai ®[1,2,3] ✉, Tingting Sui[1] ✉ & Zhanjun Li ®[1] ✉

The small size of the Cas nuclease fused with various effector domains enables a broad range of function. Although there are several ways of reducing the size of the Cas nuclease complex, no efficient or generalizable method has been demonstrated to achieve protein miniaturization. In this study, we establish an Interaction, Dynamics and Conservation (IDC) strategy for protein miniaturization and generate five compact variants of Cas13 with full RNA binding and cleavage activity comparable the wild-type enzymes based on a combination of IDC strategy and AlphaFold2. In addition, we construct an RNA base editor, mini-Vx, and a single AAV (adeno-associated virus) carrying a mini-RfxCas13d and crRNA expression cassette, which individually shows efficient conversion rate and RNA-knockdown activity. In summary, these findings highlight a feasible strategy for generating downsized CRISPR/Cas13 systems based on structure predicted by AlphaFold2, enabling targeted degradation of RNAs and RNA editing for basic research and therapeutic applications.

The CRISPR–Cas system has been developed to be a revolutionary gene-editing technology that shows precise targeting, high efficiency, and programmable DNA- or RNA-targeting properties[1,2] that has been developed for basic research, therapeutics, nucleic acid detection, diagnostics, imaging, and antiviral applications[3–11]. However, the CRISPR–Cas editing systems for the therapeutic applications, along with the promoter, expression regulatory elements and other functional proteins, remain challenging due to the in vivo delivery constraints of adeno-associated virus (AAVs)[12,13]. The small Cas enzymes have attracted considerable attention as they transcend protein size restrictions for use in broad applications, such as CasMINI[14], Cas12f[15], OMEGA[16], CasX[17,18], CasY[18], CasΦ[19], mini-dSpCas9:VPR, and mini-SaCas9:VPR[20].

Previous studies have shown several successful strategies used to miniaturize Cas proteins. An effective method is initiated by searching for natural compact Cas enzymes in metagenomic datasets, such as the 775 amino acids (AAs) Cas13X.1 and 790 AAs Cas13bt1[21,22]. However,

these compact Cas proteins do not completely recapitulate the advantages of previously optimized Cas proteins for example. Cas13bts show less efficiency than RfxCas13d for RNA transcript knockdown. In another strategy of miniaturization, the functionalities of Cas enzymes are modified. However, in this strategy, the Cas DNA-/RNA-cleavage activity is commonly eliminated, whereas its DNA-/RNA-binding activity is preserved. For example, base editors, such as HNHx-ABE and xABE, which remove a functional domain, are typically constructed by maintaining deaminase and DNA-/RNA-binding activity[15,23]. Additionally, MISER presents the overall absence of SpCas9 by generating a library of random deletion mutants, enabling Δ3CE and Δ4CE constructs to retain only approximately 72% and 63% of the primary dead SpCas9 protein sequence, respectively, with a commensurate decline in DNA-binding and cleavage activity[24]. Furthermore, although a previous study has suggested that constructing surface-localized deletion variants of poorly conserved Cas13d orthologs allow protein size reduction with minimal loss of efficiency, this process is largely

[1]State Key Laboratory for Diagnosis and Treatment of Severe Zoonotic Infectious Diseases, Key Laboratory for Zoonosis Research of the Ministry of Education, Institute of Zoonosis, and College of Veterinary Medicine, Jilin University, 130062 Changchun, China. [2]Jilin Provincial Key Laboratory of Animal Embryo Engineering, College of Veterinary Medicine, Jilin University, Changchun, China. [3]Key Laboratory of Regenerative Biology, Guangzhou Institutes of Biomedicine and Health, Chinese Academy of Sciences, 510530 Guangzhou, Guangdong, China. [4]These authors contributed equally: Feiyu Zhao, Tao Zhang. ✉ e-mail: lai_liangxue@gibh.ac.cn; suitingting@jlu.edu.cn; lizj_1998@jlu.edu.cn

restricted by the available fragments being unable to meet the miniaturization requirements of a Cas protein for future applications[25]. Because of the aforementioned challenges, it is necessary to develop a miniaturization strategy for in vivo gene therapy.

The CRISPR–Cas13 system has been described as a crRNA-mediated CRISPR nuclease that exclusively targets RNA[26–29] and is classified into six subtypes, Cas13a-d, Cas13X and Cas13Y. Functionally, all Cas13 effectors carry a pre-crRNA processing catalytic center and a target RNA cleavage catalytic center formed by two conserved R-X4-H motifs[30]. In addition, Cas13a and Cas13d both comprise a crRNA recognition (REC) lobe and a nuclease (NUC) lobe, which exhibit similar conformational changes from the binary surveillance complex to the ternary complex; for example, the binding channel is enlarged to accommodate the crRNA-target RNA duplex and the distance of the R-X4-H motif between the HEPN1 domain and HEPN2 domain is reduced to activate the HEPN nuclease active site[31–33]. Differently, the open conformation of HEPN1 and Helical-2 is the major conformational change of Cas13b from binary to ternary[34,35]. Furthermore, the results of thermal denaturation assays indicated that HEPN1 and Helical-2 assume an open conformation to allow target RNA to enter the Cas13b central cavity, initiating target recognition starting from the 3' side of the spacer sequence in PbuCas13b[35]. This open conformation was demonstrated in the Cas13bt3-crRNA-targetRNA ternary complex[36].

Thus, we propose an IDC strategy for protein miniaturization with the goal of maximizing protein miniaturization while preserving protein function based on protein structure in this study.

Interaction refers to the interaction between functional sites of Cas enzymes (Fig. S1). The Cas13 enzyme exhibits precise targeted cleavage in the form of RNP, which includes the binding of crRNA and the binding, cleavage, and release of target RNA[26–29]. Conformational degrees of freedom describe fluctuations around a native conformation and switching to functional states[37]. In the Cas13-crRNA-target RNA ternary complex, sites with an effective degree of freedom are concentrated in crRNA-target RNA contact sites, pre-crRNA processing sites, and allosteric target RNA cleavage sites[19]. Therefore, we chose the secondary structural element with the highest effective degrees of freedom as the research object for observing distribution in the relative domain.

Dynamics refers to the conformational change within a domain[38]. Upon crRNA binding and target RNA binding, domains are rearranged during the transition, resulting in a particularly noticeable change in the overall conformation of a domain; notably, the intradomain reorganization of secondary structural units is changed to a lesser extent[28,29]. The conformational variations of a domain contribute to the conformational variation in an entire protein, partly because of the long-range correlations in the native dynamics of proteins and local perturbations at any residue that can be sensed by other spatially distal residues in a protein molecule[39]. In addition, the portions of domain units are not essential to domain function; for example, HEPN domains can present morphologies that are tailored to protein function by integrating them into different units. The HEPN domain of Cas13a is larger than the normal HEPN domain because, in Cas13a, HEPN comprises numerous extra helices and a tiny hairpin insertion between α2 and α3[40]. In addition, secondary structural units with low correlations and less dramatic functional conformational change for suitability as research objects are deleted during the intradomain reorganization of the Cas13 enzyme.

Conservation refers to the conserved structural units of the Cas13d protein family, as well as the structure-specific units of a protein in the family. Attributing to the detailed prediction of the structure by AlphaFold2, conserved structural units in the Cas13d protein family and unique structural units of each Cas13d homolog are determined[41]. Although sequence homology is low, the structures of Cas13d protein family members display extraordinarily high similarity. For conserved structural units, analyzed deleted fragments are directly generalizable to the entire Cas13d protein family; for unique structural units of a particular Cas13d homolog, interaction and dynamics analyses are performed to evaluate whether they fulfilled the deletion criteria. Furthermore, the principle of fragment deletion is irrelevant to the conservation of protein sequences; even structures with highly conserved sequences can be considered for fragment deletion.

In this study, considering protein structure, we propose an IDC strategy for protein miniaturization based on the nucleoprotein interaction, dynamic conformation reorganization, and orthologs conservation. We apply the IDC strategy to generate five miniaturized variants of the Cas13 protein and verify that these ultrasmall variants of Cas13 all mediated effective mammalian transcript knockdown comparable to that of WT proteins. Furthermore, we engineer a compact RNA base editor composed of engineered ADAR and mini-RfxCas13d that exhibits robust editing efficiency relative to that of Vx and demonstrate packaging of mini-RfxCas13d within a single adeno-associated virus. Moreover, it efficiently knocks down *Pcsk9* transcript expression. In summary, by incorporating structural resolution and artificial intelligence, this strategy can be generalized to the entire Cas13 protein family.

## Results

### The design of structurally miniaturized CRISPR/Cas13 proteins via the IDC strategy

To verify the optimization of the Interaction, Dynamics and Conservation (IDC) strategy, we designed eight deleted fragments in the closely related RfxCas13d ortholog utilizing the IDC strategy (Fig. 1a). The N-terminal domain (NTD) of EsCas13d consisted of two short α-helices and a β-sandwich region composed of two antiparallel 3-stranded β-sheets. In both its binary and ternary forms, the NTD interacted with the direct repeat (DR) region of the mature crRNA, with α-helix1, β-sheets1,2 and a flexible linker connecting NTD to HEPN1 (Movie S1). β-sheet3-6 (residues ~88–124) had no contact with crRNA and away from crRNA-binding central channel, which correspond to Arg36-Lys72 of the RfxCas13d structure predicted by AlphaFold2. Thus, Arg36-Lys72 was selected as the deletion fragment of Δ1 (Fig. 1b).

The HEPN1 catalytic domain of EsCas13d was the structural scaffold between the two lobes of Cas13d and the main domain constituting the crRNA channel. The region forming the central channel in HEPN1 had extensive interactions with mature crRNA and complementary target protospacers (Movie S2). In contrast, the region locating on the protein surface (residues ~229–253) in HEPN1 was least affected by the transition from the binary complex to the ternary complex and had no role in interactions with RNA, which correspond to AlphaFold2's prediction of α-helix4,5 and β-sheet9 (Lys152-Lys251) of RfxCas13d. We chose Lys152-Lys251 as Δ2 (Fig. 1c).

The Helical-1 domain of EsCas13d underwent the most striking polypeptide rearrangement, moving outward to accommodate target RNA binding inside an enlarged cleft, though intra-domain reorganization in Helical1 was relatively little changed (Movie S3). The interaction sites were concentrated in α-helix9,10,12,13 of RfxCas13d (Fig. 1d). We picked residues ~343–364 of EsCas13d for deletion, which correspond to Δ3(Ala202-phe309) of RfxCas13d (Fig. 1d).

Portions of Helical-2 of EsCas13d showed subtle structural changes, particularly those close to the 3' end of the spacer (Movie S4), corresponding to α-helix21,22,24,25 of RfxCas13d (Fig. 1e). We considered residues ~601–628 and ~693–708 of EsCas13d for deletion, which corresponds to Δ5(Asn559-Asn579) and Δ6 (Gly655-Asp679) of RfxCas13d (Fig. 1e). The N-terminal and part of the C-terminal in HEPN2 of EsCas13d were also major components of the RNA-binding channel, corresponding to α-helix27 and 38 of RfxCas13d (Movie S5). Residues ~873–910 of EsCas13d were selected for deletion, which correspond to Δ8(Ile878-Val914) of RfxCas13d (Fig. 1f).

The comparison of structural conservation of the Cas13d orthologs showed that α-helix11 of RfxCas13d was longer than that of

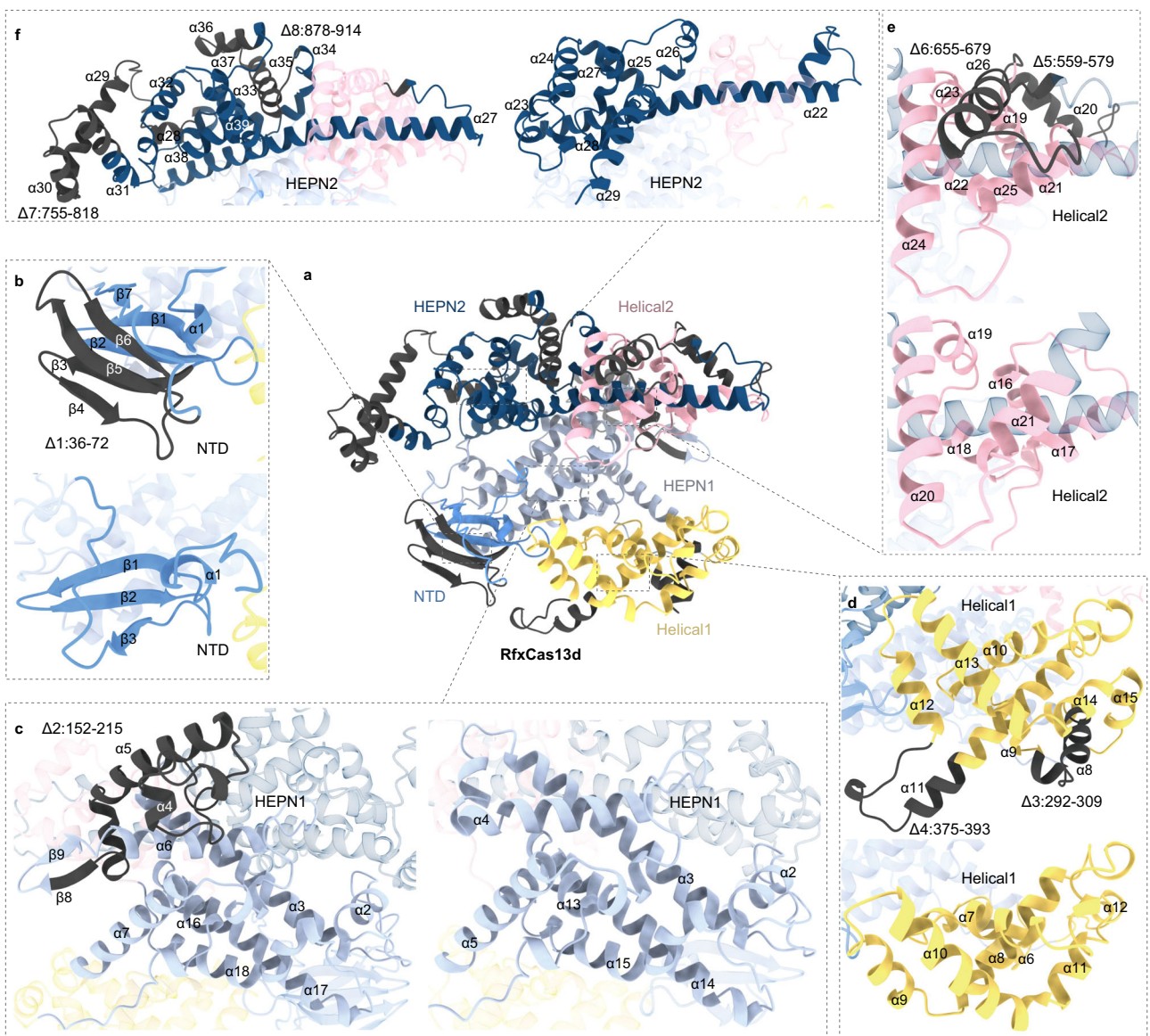

**Fig. 1 | Predicted full-length structures of RfxCas13d and mini-RfxCas13d by AlphaFold2. a** Overall structure of RfxCas13d. **b** Comparison of the NTD domains (blue) of RfxCas13d and mini-RfxCas13d. **c** Comparison of the HEPN1 domains (gray) of RfxCas13d and mini-RfxCas13d. **d** Comparison of the Helical1 domains (yellow) of RfxCas13d and mini-RfxCas13d. **e** Comparison of the Helical2 domains (pink) of RfxCas13d and mini-RfxCas13d. **f** Comparison of the HEPN2 domains (dark blue) of RfxCas13d and mini-RfxCas13d. Truncated regions is showed in dark gray.

EsCas13d in the Helical-1 domain (Figs. 1d, S10a, and S18). Due of the longer part's absence of interaction to RNA and formation of crRNA channel, Asp375-Ser393 was removed as Δ4. Similarly, RfxCas13d has two more α-helix29 and 30 than EsCas13d in HEPN2, which is close to the DR region's 5nt loop (Figs. 1f and S9b and Movie S5). Since it protruded away from protein and was exposed to the solvent, it had no interaction with Cas13d. Therefore, we deleted Lys755-Arg818 as Δ7 (Fig. 1f).

**Minimal RfxCas13d protein designed via the IDC strategy**

To evaluate the effect of RNA knockdown on the 8 mutants, we tested the efficiency at endogenous sites in HEK293 cells (Fig. 2a). The crRNA parameters of RfxCas13d, the 30-nucleotide (nt) spacer and 36-nt direct repeat (DR) sequences were selected since the deleted variant crRNA-binding channel we had generated exerted only a negligible effect on RfxCas13d[28]. The qRT-PCR results showed that all variants exhibited high knockout efficiency, with the efficiency of variants Δ4, Δ7, and Δ8 was not significantly different from that of RfxCas13d

(Fig. 2b). Moreover, the eight variants were consolidated to produce mini-RfxCas13d with only 682 AAs (Fig. 2a). Similarly, quantitative results showed that mini-RfxCas13d exhibited full activity similar to that of the wild type enzyme (Fig. 2b, c). Western blot assay showed that the protein levels of mini-RfxCas13d and RfxCas13d were similar after transfection (Fig. S2a). These results demonstrated that mini-RfxCas13d tolerated the absence of these eight segments. In addition, the internal-binding channel leading to crRNA in the five domains of mini-RfxCas13d remained intact, as demonstrated by Alpha-Fold2 prediction (Fig. 2d).

To thoroughly confirm the function and characteristics of mini-RfxCas13d, we examined the processing of pre-crRNA and the cleavage of ssRNA in vitro, respectively. We found that mini-RfxCas13d still retained similar characteristics (Fig. S3a–c). To assess whether mini-RfxCas13d is capable of autonomous pre-crRNA processing, which is similar to RfxCas13d, we purified recombinant versions of the RfxCas13d and mini-RfxCas13d effectors (Fig. S4a, b). The effectors were then incubated with transcribed pre-crRNAs in vitro consisting of

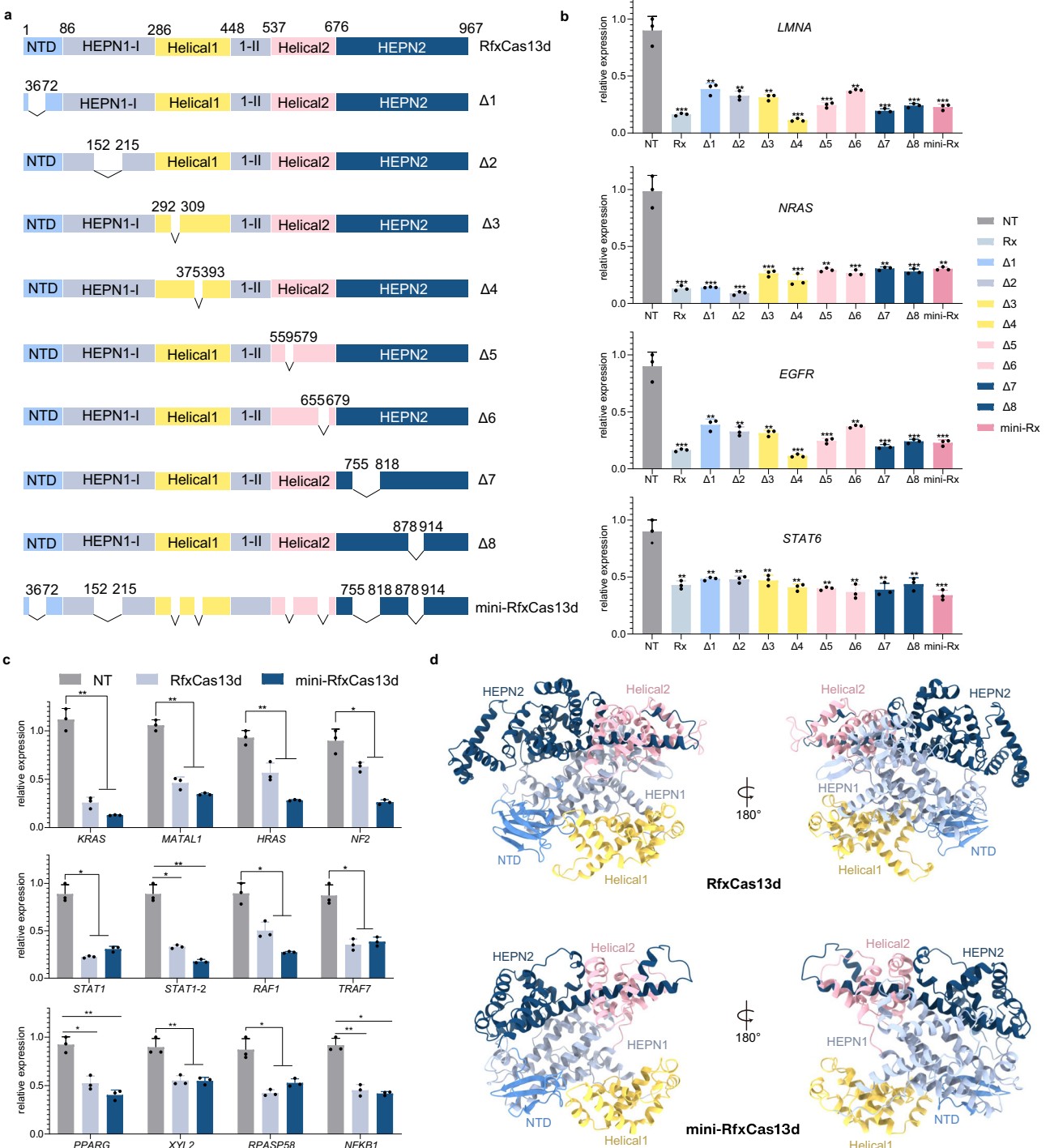

**Fig. 2 | RfxCas13d tolerates multiple domain deletions while maintaining target-binding activity and RNA-knockdown activity. a** Schematic showing the domain organization of the RfxCas13d, mini-RfxCas13d and Δ1-Δ8 deletion variants. **b** Comparison of the knockdown efficiency for endogenous transcripts by RfxCas13d, mini-RfxCas13d and Δ1-Δ8 in HEK293T cells. NT, non-target crRNA. The data are presented as the mean ± SD. Two-tailed unpaired two-sample *t* test. *$P < 0.05$, **$P < 0.01$, ***$P < 0.001$ (*n* = 3 biological replicates, each with an average of 4 technical replicates). Source data are provided as a Source date file. Rx:

RfxCas13d, mini-Rx: mini-RfxCas13d. **c** Comparison of the knockdown efficiency for 12 endogenous transcripts by RfxCas13d and mini-RfxCas13d. NT, non-target crRNA. The data are presented as the mean ± SD. Two-way analysis of variance (ANOVA) followed by Dunnett's multiple comparisons. *$P < 0.05$, **$P < 0.01$, ***$P < 0.001$. (*n* = 3 biological replicates, each with an average of 4 technical replicates). Source data are provided as a Source data file. **d** The overall predicted structure of RfxCas13d and mini-RfxCas13d by AlphaFold2 are shown in two different orientations and color coded as indicated in (**a**).

the repeat-spacer-repeat-spacer construction. We found that RfxCas13d and mini-RfxCas13d effectors similarly process pre-crRNAs to form mature crRNAs (Fig. S4e). We next targeted RfxCas13d and mini-RfxCas13d enzyme-crRNA complexes to ssRNA substrates containing target sequences complementary to the crRNA spacer and

demonstrated targeted RNA cleavage activity for both enzymes (Fig. S4g).

To examine whether mini-RfxCas13d tolerates mismatches, we introduce single mismatches into spacer sequences. For the *KRAS* locus, mini-RfxCas13d and hfRfxCas13d[42], which are comparable to

RfxCas13d, showed limited tolerance to crRNA at the 3′ end and high tolerance to crRNA at the 5′ end (Fig. S5a). Previous study also had shown that seed regions for RfxCas13d knockdown had been identified between crRNA nucleotides 15–21, with the core at nucleotide 18 relative to the crRNA 5′-end[43]. To identify the optimal spacer length for mini-RfxCas13d relative to RfxCas13d and hfRfxCas13d, we designed different length spacers ranging from 15 to 50nt targeting the endogenous *KRAS* locus. The results showed that crRNAs at 30nt present the strongest target knockdown with all test Cas13ds (Fig. S5b). Furthermore, we performed transcriptome-wide RNA-sequencing (RNA-seq) analysis on mini-RfxCas13d targeting *B4GALNT1* and *NF2* genes in HEK293T cells. It was found that mini-RfxCas13d induced less differentially expressed genes than RfxCas13d and more than hfRfxCas13d after knockdown of *NF2* (Fig. S6a–c). We introduced the mutations of N2V8 variants into mini-RfxCas13d to generate mini-hfRfxCas13d. The transcriptome-wide RNA-sequencing (RNA-seq) analysis showed that mini-hfRfxCas13d exhibited more differentially expressed genes than hfRfxCas13d after knockdown of *NF2* (Fig. S6d). In addition, mini-RfxCas13d exhibited more differentially expressed genes than hfRfxCas13d after knockdown of *B4GALNT1* (Fig. S6e, f).

In conclusion, mini-RfxCas13d efficiently mediates crRNA-guided knockdown of endogenous sites with little loss of efficiency and exhibits similar characteristics in optimal spacer length, mismatch tolerance, and off-targets in the transcriptome compared to that of RfxCas13d.

## Miniaturization of the CRISPR/Cas13d family by Alpha-Fold2 via the IDC strategy

Analyzing the structure of proteins in a family remains challenging due to the small structural coverage fraction. To overcome this limitation, we used AlphaFold2 to predict protein structures with atomic accuracy. Specifically, to observe the structural conservation of Cas13d family proteins, Alpha-Fold2 was used to predict the structure of RspCas13d and RfxCas13d. The results showed that RspCas13d and RfxCas13d exhibited high similarity with EsCas13d (Figs. 2c, S7a, and S8a). However, RfxCas13d carried a large insertion in the Helical1 and HEPN2 domains, in contrast to RspCas13d and EsCas13d (Figs. S9b and S10a). Thus, to verify that the IDC strategy is sufficient as a general strategy for Cas13d protein miniaturization, we designed to miniaturize the RspCas13d and EsCas13d by deleting the part of all domains including NTD, HEPN1, HEPN2, Helical1 and Helical2 (Figs. 3b, S9, and S10). Eventually, we generated mini-RspCas13d and mini-EsCas13d based on their high similarity (Figs. 3c, d, S7b, and S8b). We assessed the degree of RNA knockdown mediated by two mini-deletion variants using crRNA arrays of the Cas13 enzyme from natural microorganisms[29]. The quantitative results showed that mini-EsCas13d and mini-RspCas13d slightly outperformed EsCas13d and RspCas13d, respectively (Fig. 3e, f). Therefore, the IDC strategy of miniaturizing structurally similar homologous proteins can be applied to the Cas13d family through structure prediction using Alpha-Fold2.

## Miniaturization of the CRISPR/Cas13b family by Alpha-Fold2 via the IDC strategy

The Cas13b protein is more appropriate for in vivo application to mammalian systems than the Cas13d protein due to the lower collateral effects of the former[44]. Analysis of the predicted structure revealed that PspCas13b shared high structural homology with the resolved PbuCas13b (Figs. 4, 5a, b, S11, and S12). To test the generalizability of the IDC strategy, we developed mini-PspCas13b and mini-PbuCas13b (Figs. 5a, b, S11, and S12).

To assess whether mini-Cas13bs are capable of autonomous pre-crRNA processing and ssRNA cleavage activity, we purified the recombinant version of the PspCas13b and mini-PspCas13b effectors (Fig. S4c, d). We next examined the pre-crRNA cleavage and RNA-guided ssRNA cleavage activity of the mini-PspCas13b enzyme in vitro. We observed that mini-PspCas13b still possesses the similar activity of processing pre-crRNA and cleaving target ssRNA efficiently as PspCas13b (Fig. S4f, h). To examine whether the efficiency of RNA knockdown at endogenous sites was influenced by miniaturization, the western blot assays were performed to verify the protein level at first. As shown in Fig. S2b, the results showed that the protein levels of PbuCas13b, and mini-PbuCas13b were similar after transfection. Furthermore, we performed quantitative reverse transcription PCR (RT-qPCR) analysis of gene expression. As shown in Fig. 5e, f, the expression of the genes mini-PspCas13b and mini-PbuCas13b was not significantly reduced compared with that of the wild type proteins. Moreover, the two mini variants exhibited the same degree of targeting RNA degradation as their wild type Cas13b counterparts, indicating that the 12 deleted segments exerted little effect on Cas13b functioning. Therefore, the IDC strategy of miniaturizing structurally similar homologous proteins is also applicable to the Cas13b family after structure prediction by Alpha-Fold2.

## Precise RNA editing via dmini-RfxCas13d-ADAR2DD-E488Q fusion

To develop a mini-RfxCas13d protein for RNA editing, we fused the catalytically inactive mini-RfxCas13d with a highly active mutant of the ADAR2 catalytic domain in RNA adenosine deaminase (ADAR2DD-E488Q), thus establishing dmini-RfxCas13d-ADAR (Fig. 6a). Huang fused ADAR2DD-E488Q and APOBEC3A deaminase between AA 558 and AA 559 of RfxCas13d to generate a highly efficient RNA A-I base editor, Vx, and RNA C-U base editor, CURE-X[45,46]. Similarly, ADAR2DD-E488Q was fused to Helical2 in mini-RfxCas13d between AA 420 and AA 421. The ability of dmini-RfxCas13d-ADAR to introduce A-I mutations was evaluated in human 293 cells. Sanger sequencing showed that the A-to-I conversion rate of dmini-RfxCas13d-ADAR was comparable to that of dRfxCas13d-ADAR (Figs. 6b and S13). To assess the off-target edits at the base level, we performed next-generation sequencing of the mutant sites, and the results showed a slight reduction in the off-target effect of dmini-RfxCas13d-ADAR relative to that of dRfxCas13d-ADAR (Fig. 6c, d). As expected, the off-target edit sites of in transcriptome-wide by dmini-RfxCas13d-ADAR slightly reduced relative to dRfxCas13d-ADAR at endogenous *COG3* locus by RNA-seq analysis (Fig. S14). Overall, dmini-RfxCas13d-ADAR led to efficient crRNA-dependent A-to-I conversions at mismatched base positions. Together, these results showed that the IDC strategy demonstrates great potential for use in generating compact and efficient RNA base editors that facilitate an AAV-based method for treating genetic diseases.

## Reduction of serum cholesterol after mini-RfxCas13d-mediated knockdown of *Pcsk9* in the liver

AAV delivery of RfxCas13d to mice has been widely used for accurate targeted therapy in preclinical studies[47, 48]. To determine whether the efficiency of mini-RfxCas13d-mediated knockdown in vivo, a regulatory metabolism gene, the murine proprotein convertase subtilisin/kexin type 9 (*Pcsk9*) gene, was chosen for our investigation. Specifically, we evaluated whether expression of *Pcsk9* was suppressed by determining whether serum cholesterol levels were restored[49]. A previous study had demonstrated that the RfxCas13d system efficiently targeted RNA for knockdown in hepatocytes in vivo. Therefore, we first designed three crRNAs to target the coding sequence in *Pcsk9* mRNA, as they had downregulated the level of *Pcsk9* mRNA in a previous study (Table S5). Plasmid carrying mini-RfxCas13d and each *Pcsk9* crRNA were transfected into mouse neuroblastoma (N2a) cells that overexpressed a *Pcsk9* mRNA fragment. As expected, all *Pcsk9* crRNA candidates convincingly knocked down the *Pcsk9* mRNA (Fig. S15). After knocking down a gene in cultured mammalian cells in vitro, the feasibility of in vivo metabolic gene knockdown was assessed. First, we

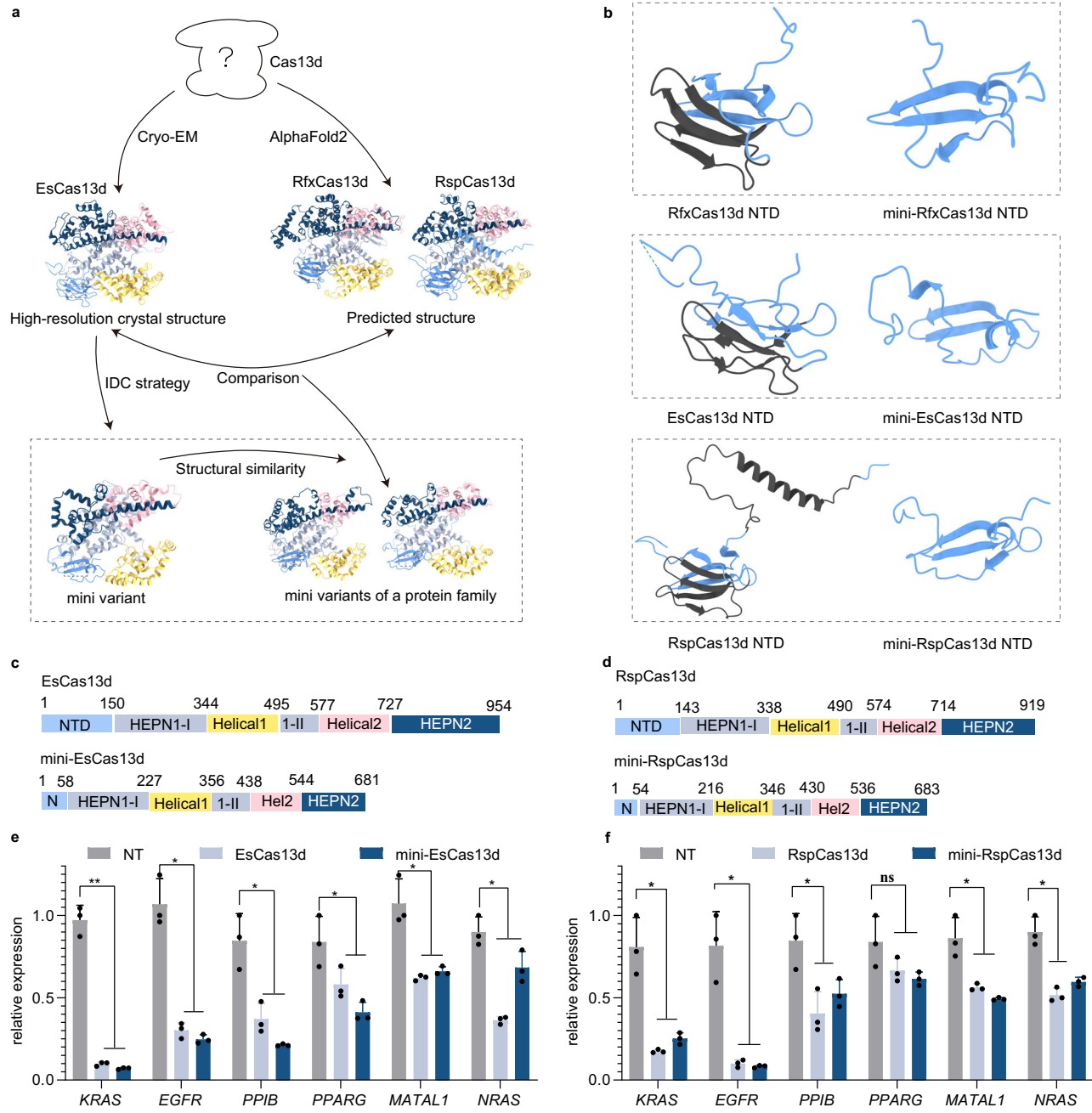

**Fig. 3 | Cas13d orthologs tolerate multiple domain deletions while maintaining target-binding and RNA-knockdown activity. a** Schematic diagram showing the miniaturization of EsCas13d to Cas13d orthologs using the IDC strategy and AlphaFold2. **b** Comparison of the NTD domains of RfxCas13d and mini-RfxCas13d, EsCas13d and mini-EsCas13d, RspCas13d and mini-RspCas13d. RfxCas13d, mini-RfxCas13d, RspCas13d, mini-RspCas13d and mini-EsCas13d were predicted by AlphaFold2, except for EsCas13d (6E9E). Deletion regions were colored deep gray. **c** Schematic showing the domain organization of EsCas13d and mini-EsCas13d. **d** Schematic showing the domain organization of RspCas13d and mini-RspCas13d. **e** Comparison of the EsCas13d and mini-EsCas13d knockdown efficiency of endogenous transcripts. The data are presented as the mean ± SD. Two-way analysis of variance (ANOVA) followed by Dunnett's multiple comparisons. *$P < 0.05$, **$P < 0.01$. ($n = 3$ biological replicates, each with an average of 4 technical replicates). Source data are provided as a Source data file. **f** Comparison of the endogenous transcripts knockdown efficiency of RspCas13d and mini-RspCas13d. NT non-target crRNA. The data are presented as the mean ± SD. Two-way analysis of variance (ANOVA) followed by Dunnett's multiple comparisons. *$P < 0.05$, **$P < 0.01$, ns, not significant. ($n = 3$ biological replicates, each with an average of 4 technical replicates). Source data are provided as a Source data file.

constructed AAV-CMV-mini-RfxCas13d (with crRNA2 targeting *Pcsk9*) driven by the CMV promoter to knock down *Pcsk9* specifically in the liver. The construct was packaged into AAV8 and injected into the tail vein of 8-week-old mice (Figs. 6e and S15).

After injection, the *Pcsk9* mRNA levels in AAV8-injected mice were significantly decreased, to 34% ± 3.8% of those in the non-injected mice 3 weeks after AAV injection (Fig. 6f), and the *Psck9*

protein levels in the AAV8-injected mice were also knocked down (Fig. 6g). The total serum cholesterol level in the *Pcsk9* knockdown mice was reduced to 61% ± 8.3% that of the normal level (Fig. 6h). Moreover, serum alanine aminotransferase (ALT) and AST levels were not significantly changed in the AAV8-injected mice, suggesting that the treatment did not induce toxicity or liver damage (Fig. 6i, j). Furthermore, histological analysis did not reveal any significant

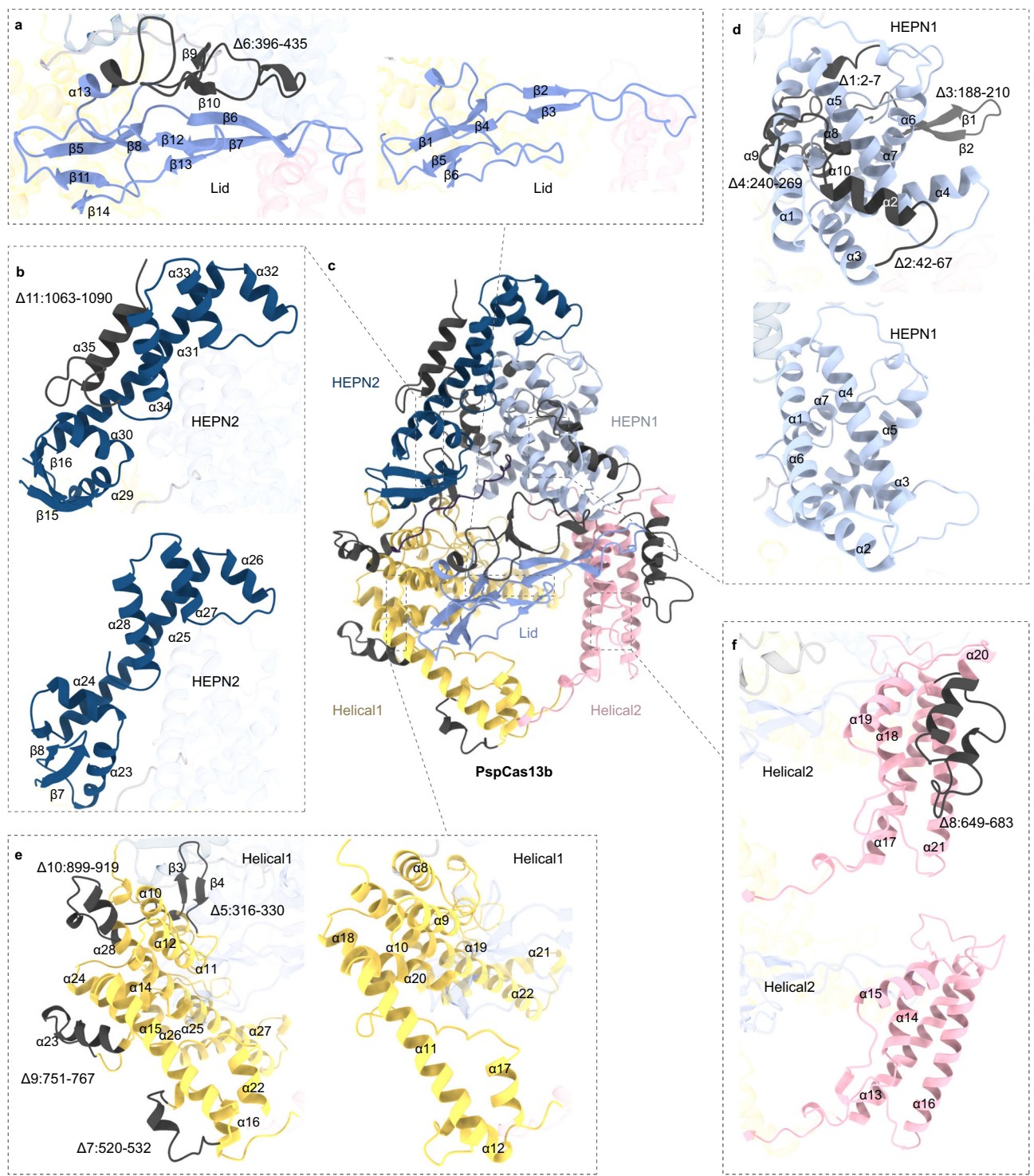

**Fig. 4 | Predicted full-length structures of PspCas13b and mini-PspCas13b by AlphaFold2. a** Comparison of the Lid domains (purple) of PspCas13b and mini-PspCas13b. **b** Comparison of the HEPN2 domains (dark blue) of PspCas13b and mini-PspCas13b. **c** Overall structure of PspCas13b. **d** Comparison of the HEPN1 domains (gray) of PspCas13b and mini-PspCas13b. **e** Comparison of the Helical1 domains (yellow) of PspCas13b and mini-PspCas13b. **f** Comparison of the Helical2 domains (pink) of PspCas13b and mini-PspCas13b. All deletion regions were colored deep gray.

differences in hepatocyte features or the intact hepatic lobule structure between AAV8-injected mice and noninjected mice (Fig. S15). These results confirmed that mini-RfxCas13d is an effective tool for knocking down *Pcsk9* mRNA, which modulated the cholesterol levels in mice in vivo. Therefore, the mini-RfxCas13d demonstrates great potential to be a compact and efficient tool to facilitate AAV-based treatments of genetic diseases.

## Discussion

In this study, we proposed an IDC strategy and engineered five mini variants of Cas13b and Cas13d that retained both full RNA-binding activity and full RNA-cleavage activity. Using this IDC strategy and AlphaFold2-based structural information, we describe a strategy for protein miniaturization. Moreover, the function of the mini variants was ascertained, and the mini variants showed performances

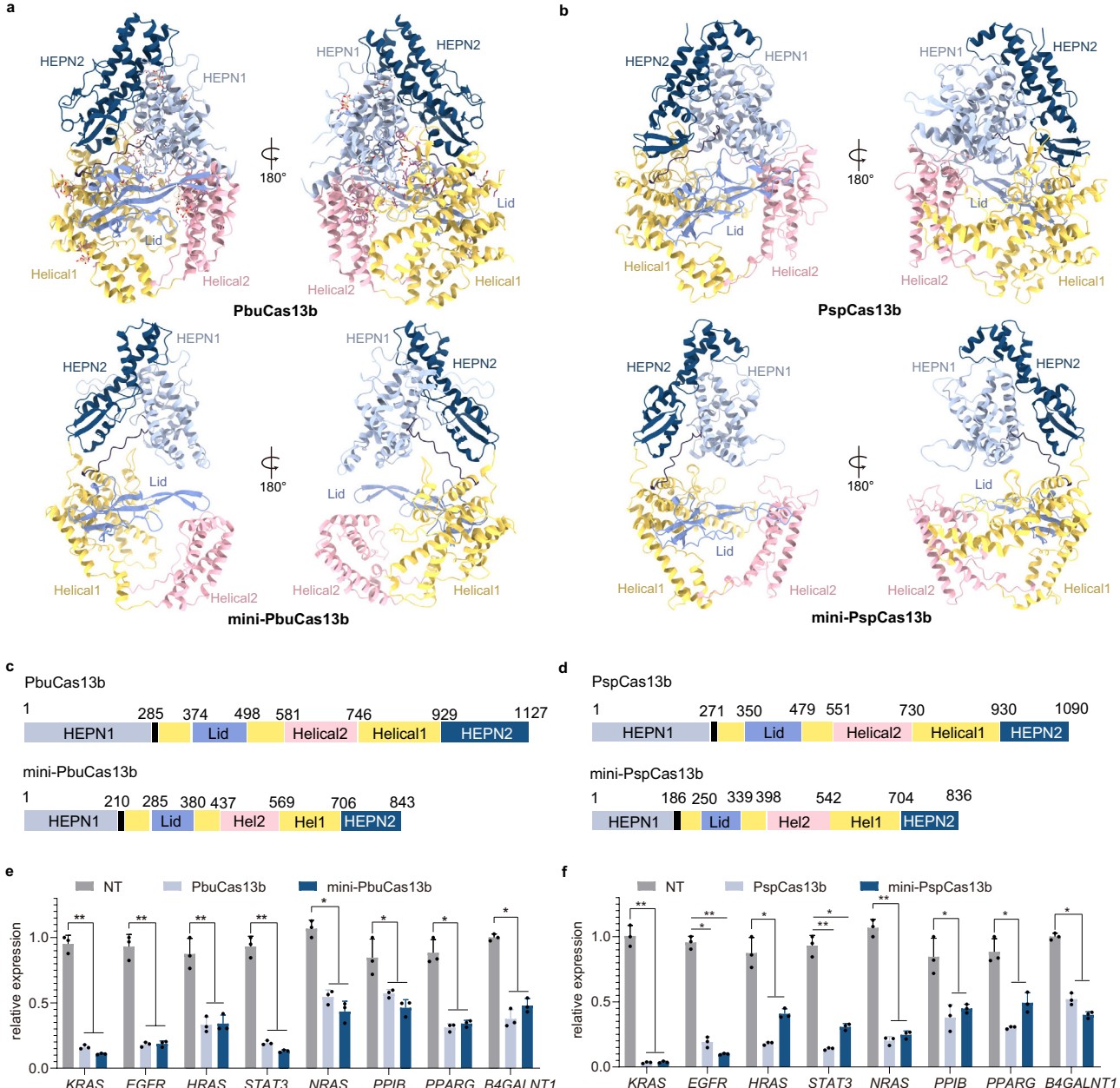

**Fig. 5 | PspCas13b tolerates multiple domain deletions while maintaining target-binding and RNA knockdown activity. a** The cryo-electron microscopy structure of PbuCas13b (6DTD) and the predicted overall structure of mini-PbuCas13b by AlphaFold2 are shown in two different orientations. **b** The predicted overall structure of PspCas13b and mini-PspCas13b by AlphaFold2 are shown in two different orientations. **c** Schematic showing the domain organization of PbuCas13b and mini-PbuCas13b. **d** Schematic domain showing the organization of PspCas13b and mini-PspCas13b. **e** Comparison of endogenous transcript knockdown efficiency of PbuCas13b and mini-PbuCas13b. The data are presented as the mean ± SD.

Two-way analysis of variance (ANOVA) followed by Dunnett's multiple comparisons. **$P < 0.01$. ($n = 3$ biological replicates, each with an average of 4 technical replicates). Source data are provided as a Source Date file. **f** Comparison of endogenous transcript knockdown efficiency of PspCas13b and mini-PspCas13b. NT nontarget crRNA. The data are presented as the mean ± SD. Two-way analysis of variance (ANOVA) followed by Dunnett's multiple comparisons. *$P < 0.05$, **$P < 0.01$. ($n = 3$ biological replicates, each with an average of 4 technical replicates). Source data are provided as a Source data file.

comparable to those of the WT protein in base-editing and in vivo experiments, suggesting a method for structure analysis and a use for AlphaFold2.

The IDC strategy offers many advantages compared with other methods of miniaturization. A multidimensional analysis of proteins preserved the RNA-binding and cleavage activity of the mini-Cas13 enzymes with little efficiency loss compared to that of the wild type Cas13 enzyme. The use of functional regions, in contrast to those described in the literature, enabled them to be directly studied as research objects, in contrast to studies based on changes in

functionalities, such as the analysis of HNHx-ABE and xABE4[15,20,23]. The selection of deleted fragments was based on protein structure, not the primary sequence, such as the N or C terminal domain of a protein[15,20,23]. Moreover, the mini-Cas13 enzyme retained only approximately 70% of the primary sequence of the wild-type Cas13 enzyme, permitting the fusion of various effector domains, such as the ADAR2 catalytic domain[7]. The median size of the mini-Cas13b and mini-Cas13d proteins was 250 and 270 amino acids smaller than that of Cas13b and Cas13d, respectively. In addition, the IDC strategy was shown to be a generalized method for Cas enzyme miniaturization.

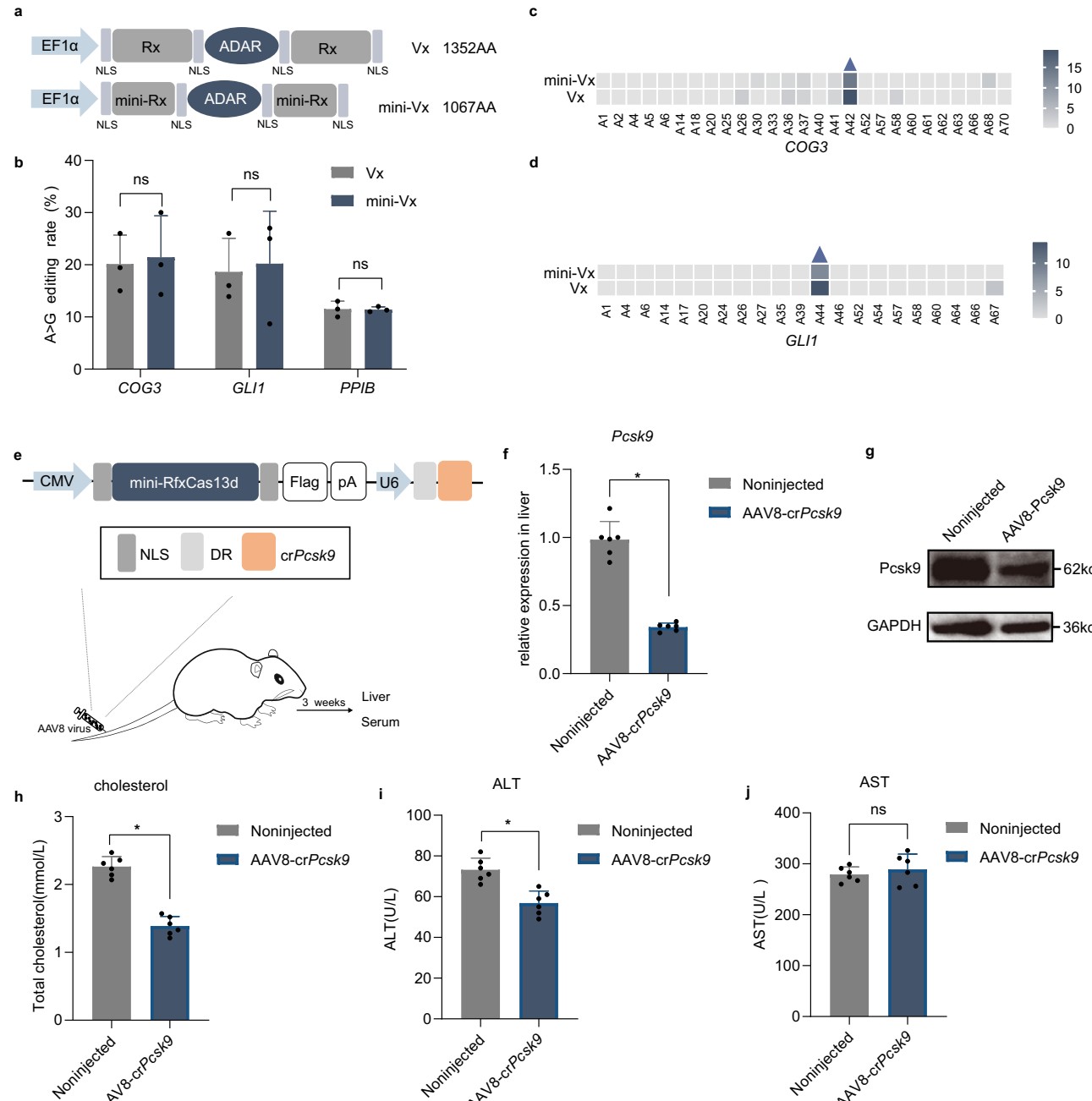

**Fig. 6 | Applications including base editing and modulation of the mini variants in vivo. a** Schematic representation of Vx and mini-Vx. Vx: REPAIRx, mini-Vx: mini-REPAIRx. **b** A-to-I editing efficiency of Vx and mini-Vx with endogenous transcripts in HEK293T cells, as analyzed by deep sequencing. The data are presented as the mean ± SD. Two-way analysis of variance (ANOVA) followed by Sidak's multiple comparisons. ns not significant. ($n = 3$ biological replicates). Source data are provided as a Source data file. **c** Quantification of the percentage *COG3* edited at the targeted adenosine (blue triangle) and neighboring sites by Vx and mini-Vx. Source data are provided as a Source data file. **d** Quantification of the percent of *GLI1* editing at the targeted adenosine (blue triangle) and neighboring sites by Vx and mini-Vx. Source data are provided as a Source data file. **e** Experimental scheme. **f** Quantification of *Pcsk9* mRNA levels in the livers from AAV8-injected mice ($n = 6$) and noninjected mice ($n = 6$). The data are presented as the mean ± SD. *P* values are by Wilcoxon matched-pairs signed-rank test. *$P < 0.05$. Source data are provided as a Source data file. **g** Quantification of *Pcsk9* protein levels in the livers from AAV8-injected mice. **h** Quantification of total serum cholesterol levels at 3 weeks (noninjected, $n = 6$; crPcsk9, $n = 6$). **i** Serum ALT was quantified in noninjected mice ($n = 6$) and AAV8-crPcsk9-injected mice ($n = 6$). **j** Serum AST was quantified in noninjected mice ($n = 6$) and AAV8-crPcsk9-injected mice ($n = 6$). **h**–**j** The data are presented as the mean ± SD. *P* values are by Wilcoxon matched-pairs signed-rank test. *$P < 0.05$, ns not significant. Source data are provided as a Source data file.

The mini variant of RfxCas13d could be generalized into other mini variants of Cas13d orthologs by comparing the structure predicted by AlphaFold2, as Cas13d proteins present with very similar structures. Furthermore, "interactions" in the IDC strategy were appropriate for analyzing proteins that collaborate with nucleic acids. Therefore, the IDC strategy can theoretically be used for the entire Cas enzyme family. This feature provided a theoretical foundation for the miniaturization of other Cas enzyme isoforms. Finally, except for strategies that include deleted fragments, the developed optimization strategy can be used with miniaturized variants. For instance, REPAIRx (Vx), which is a combination of RfxCas13d and a deaminase domain inserted into the middle, is precise yet highly efficient, outperforming various

previous versions with both mRNA and nuclear RNA targets[36]. Mini-Vx, a combination of mini-RfxCas13d and a deaminase domain inserted in the middle, was not only small enough to fit within an AAV vector but was also highly efficient.

The IDC strategy is not based on the intersection of three IDC factors; in contrast, it is used to evaluate whether protein function is affected, which is critical to the analysis. The HEPN2 domain of RfxCas13d, for example, accommodates a larger insertion than RspCas13d and EsCas13d, which are located adjacent to the crRNA 5′ terminal DR region. The DR region of the crRNA 5′ terminus was exposed in the solvent, as determined during the structural analysis of EsCas13d, and exhibited no interaction with the protein[19]. Therefore, this region was deduced to have exerted no effect on protein function and was therefore considered a deleted fragment.

The IDC strategy relies on precise and comprehensive structural resolution, particularly the conformational dynamics of the protein domain. Thermal denaturation that was initially used to deduce the structural changes in the ternary complex after formation complicates the analysis due to the lack of specific information on the structure of the spacer region in the binary complex of PbuCas13b and the ternary complex[29]. Additionally, mini-RfxCas13d showed less collateral cleavage activity relative to RfxCas13d (Fig. S6). Since the N3V7 fragment in hf RfxCas13d coincided with the deleted fragment of the Δ2 variants, we reasoned that the deletion of N3V7 sequence cause less collateral cleavage activity[42] (Fig. S16). Mini-hfRfxCas13d showed comparable collateral cleavage activity relative to mini-RfxCas13d. We speculated that the deletion of N3V7 or other fragments may have an impact on the function of N2V8 variant and cause diminished performance in high fidelity relative to N2V8 variants.

In summary, we demonstrated the IDC strategy for Cas13d miniaturization and Cas13b miniaturization. Mini-Cas13 fused with various effector domains shows many applications, including nucleic acid detection, imaging, splicing modification, diagnostics, therapeutics, and epitranscriptomic modifications. The IDC strategy provides an interesting viewpoint on protein miniaturization based on structural analysis and establishes a generalized miniaturization method in conjunction with AlphaFold2.

## Methods

### Ethical statement

Institute for Cancer Research (ICR) mice were obtained from the Laboratory Animal Center of Jilin University (Changchun, China). All animal studies were conducted under the guidance of the Animal Welfare and Research Ethics Committee at Jilin University (IACUC number: SY202204102).

### IDC strategy

IDC strategy workflow of Cas13 is shown in Fig. S1. The primary functions of the Cas protein are pre-crRNA processing, crRNA recognition, and target RNA cleavage. We extract the nucleoprotein interaction sites (containing pre-crRNA processing, crRNA recognition, and target RNA cleaving) in different states (binary and ternary complex) and the conformational changes from the article of EsCas13d Eyro-EM structure. Eyro-EM structures are restricted to instantaneous interactions, therefore protein interactions during the entire dynamic process must be analyzed to identify the interaction sites thoroughly. Mark the nucleoprotein interaction structural units and skeleton of the interaction channel to exclude the core region. Through previous analysis, we find that all five domains were engaged in the interaction with RNA. Nevertheless, only α1, β1, β2, and β7 interact with RNA in the domain of NTD (Fig. 1b). Additionally, the crack between the NTD and Helical1 expands when the target RNA binding, and the displacement change of β3-6 is caused by the outward movement of the NTD domain, and β3-6 is not the skeleton of the interaction channel. So, we choose β3-6 as Δ1. The same type appears in HEPN1 Δ2; Helical1 Δ3; Helical2 Δ5 and Δ6;

and HEPN2 Δ8. These 6 deleted fragments are relatively conserved in the Cas13d family protein, and can also be used as deleted fragments of EsCas13d and RspCas13d according to Figs. S17 and S18. In contrast, specific structural units of RfxCas13d (α29, α30, α31, and the lower portion of α11) are reanalyzed. For the lower portion of α11, which is not involved in interacting with RNA, movements also happen passively with Helical1, so we choose the lower portion of α11 as Δ4 (Fig. 1d); For α29, α30, and α31, it is the binding region of DR (nt 10−14) spatially, but DR (nt 10−14) is exposed to solvents and does not interact with Cas proteins, so we choose α29, 30, and 31 as Δ7(Fig. 1f). The lengthy α-helix in the N-terminal of EsCas13d and RspCas13d is their distinctive structural feature and it doesn't interact with RNA or form the skeleton of the interaction channel (Fig. 3b). We choose the lengthy α-helix in the N-terminal as a deletion fragment to generate mini-EsCas13d and mini-RspCas13d.

### Animals

Institute for Cancer Research (ICR) mice were obtained from the Laboratory Animal Center of Jilin University (Changchun, China)[50]. All animal studies were performed in accordance with the guide of the Animal Welfare and Research Ethics Committee at Jilin University.

### AlphaFold2 prediction

The whole structure of the Cas13d and Cas13b proteins was predicted by AlphaFold2 with Google Colab and default settings[34,51]. We compared the top five ranked outputs and selected Rank 1 to prepare the figures. Structural figures were generated with UCSF ChimeraX[52].

### Plasmid construction

Cas13bt, EsCas13d, RspCas13d and PspCas13b were obtained from Addgene (#176316, #108303, #108305, and #103862, respectively)[7,16,23]. EsCas13d, RspCas13d, and PspCas13b sequences were amplified by PCR and cloned into a pcDNA3.1 backbone. Cas13X (#171379) and REPAIRx were gifts from Professor Hui Yang and Xingxu Huang, respectively[15,38]. Mini-RfxCas13d, mini-EsCas13d, mini-RspCas13d, mini-PspCas13b, mini-PbuCas13b, PbuCas13b and mini-Vx were synthesized and cloned into pcDNA3.1(+) (GenScript Biotech, China). The Δ1, Δ2, and Δ3 RfxCas13d strains were constructed by homologous recombination. Δ4, Δ5, Δ6, Δ7, and Δ8 were synthesized and cloned into pcDNA3.1(+) by GenScript Biotech (China). The sequences of the proteins are listed in Table S1. All of the primers used for plasmid construction are listed in Table S2.

### Cell culture and transfection

HEK293T (ATCC) and N2A cell lines were cultured in Dulbecco's modified Eagle's medium (DMEM) supplemented with 10% fetal bovine serum (HyClone) and incubated at 37 °C in 5% $CO_2$ in 6-well poly-D-lysine-coated plates (Corning). The cells were cotransfected with Cas13 (1500 ng) and crRNA plasmids (1500 ng) using Hieff TransTM Liposomal Transfection Reagent (YEASEN, China) according to the manufacturer's instructions, and the cells were harvested 48 h later using TRIzol Universal Reagent (TIANGEN Biotech, China). All Cas13 crRNAs used in this work are listed in Table S3.

### RNA knockdown efficiency evaluation

RNA was extracted from successfully transfected cells following the description in a previous study[53]. First-strand cDNA was synthesized using a FastKing RT kit (with gDNase) (TIANGEN Biotech, China). Quantitative PCR (qPCR) was performed using QuantStudio3 (Thermo Fisher Scientific) with sample cDNA and SuperReal PreMix Plus (SYBR Green) (TIANGEN Biotech, China). The primers are shown in Table S2. The $2^{-\Delta\Delta CT}$ method was used to analyze gene expression, and GAPDH was the reference gene. All experiments were repeated three times for each gene.

## Proteins' expression and purification

The expression and purification of proteins were performed as described previously[28]. Cas13d proteins were cloned into pET-28a with both N-terminal and C-terminal 6×His. The plasmids were transformed into Rosetta2(DE3) cells (Tolo Biotech), induced with 200 μM IPTG at OD600 0.5, and grown for 16 h at 16 °C. Cells were harvested by PBS supplemented with 1× protease inhibitor solution (Solarbio) and then sonicated and clarified via centrifugation (12,000 × $g$ for 40 min at 4 °C), filtered with 0.45 μM PVDF filter and incubated with 1 mL of BeyoGold™ His-tag Purification Resin (Beyotime Biotechnology) overnight (at 4 °C). The column was balanced 2-3 times with 1 column volume of lysis buffer (50 mM Tris; 500 mM NaCl) before loading. The sample was applied to an affinity chromatography column, washed 5 times with 1 column volume of washing buffer (50 mM Tris; 500 mM NaCl; 10–20 mM imidazole) and 5 times with 1 column volume of Elution Buffer (50 mM Tris; 500 mM NaCl;250 mM imidazole). Purified, eluted proteins were stored at 4 mg/mL in Protein Storage Buffer (50 mM Tris-HCl, 1 M NaCl, 10% glycerol, 2 mM DTT).

## RNA purification and transcription

Oligonucleotide sequences of crRNA, pre-crRNA, and ssRNA were synthesized (Sangon Biotech), annealed, PCR amplified and then subsequently purified respectively. The purified templates were in vitro transcribed with the Hiscribe T7 Quick High Yield RNA Synthesis Kit (New England Biolabs) and frozen at −80 °C.

## Biochemistry cleavage reactions in vitro

The biochemistry cleavage reactions in vitro were performed as described previously[28]. Briefly, Purified RfxCas13d, mini-RfxCas13d, PspCas13b, mini-PspCas13b proteins, and crRNA or pre-crRNA were mixed at a 1:2 molar ratio in RNA Cleavage Buffer (25 mM Tris pH 7.5, 15 mM Tris pH 7.0, 1 mM DTT, 6 mM MgCl2). Before adding the target ssRNA at a 1:2 molar ratio in relation to the Cas13d proteins, the reaction was ready on ice and incubated at 37 °C for 15 min. After 45 min of 37 °C incubation, the reaction was stopped at 37 °C for 15 min with 1 L of an enzyme inhibitor solution (10 mg/mL Proteinase K, 4 M urea, 80 mM EDTA, and 20 mM Tris pH 8.0) .After the reaction was denatured at 65 °C for 10 min with 2X RNA loading buffer (13 mM Ficoll, 8 mM Urea, 25 mM EDTA), the sample was separated on a 15% TBE-Urea gel.

## RNA-base-editing efficiency evaluation

Transfected cells were sorted for RNA extraction to assess the A-to-I base-editing efficiency of mini-Vx and Vx. A FastKing RT kit (with gDNase) (TIANGEN Biotech, China) was used to reverse transcribe complementary cDNAs from RNAs, and crRNA target sites were amplified from cDNAs using 2*Taq PCR Master Mix II (TIANGEN Biotech, China) for Sanger or deep sequencing. The Sanger sequencing results were analyzed with EditR to measure the effectiveness of the A-to-I conversion at each target site (http://baseeditr.com/)[54]. Deep sequencing was performed by using Hi-TOM (http://www.hi-tom.net/hi-tom/index-CH.php)[55]. All Cas13 crRNAs used in this work are listed in Tables S4 and S5. The primers are shown in Table S2.

## RNA-seq and analysis

Forty-eight hours after transfection, total RNA was extracted from HEK293 cells. Stranded mRNA libraries were constructed by Genewiz and Novogene, and loaded on an Illumina HiSeq/Illumina Novaseq/MGI2000 instrument for sequencing using a 2 × 150 paired-end (PE) configuration according to manufacturer's instructions. Fastq passes filter data were converted to high quality, clean data using Cutadapt (V1.9.1, phred cutoff: 20, error rate: 0.1, adapter overlap: 1 bp, min. length: 75, proportion of $N$: 0.1). Using the program Hisat2 (v2.0.1), data were aligned to the reference genome. The DESeq2 Bioconductor package was utilized for the analysis of differential

expression. Data-driven prior distributions are incorporated into the estimations of dispersion and logarithmic fold changes, and the Padj of the genes was adjusted to 0.05 to identify those that were differentially expressed.

## AAV injection

PackGene Biotech (China) created the AAV-mini-RfxCas13d-*Pcsk9*-crRNA2 expression cassettes and produced the viruses (https://www.packgene.cn/about-us/).

An AAV injection assay was performed as previously described[44]. Production of AAV8-RfxCas13d vectors was performed by PackGene Biotech (China). The AAV-mini-RfxCas13d-*Pcsk9*-crRNA2 titer was $2 \times 10^{12}$ genome copies (GCs)/mL. Before injection, all AAV doses were adjusted to a 100-μL volume with sterile PBS. For AAV8 vector injections, 8-week-old male ICR mice were given $2 \times 10^{11}$ GCs per mouse intravenously via lateral tail vein injection. Twenty-one days after the injection, the mice were fasted for 12 h before liver and blood samples were collected. Housing conditions:12 h light/12 h dark, 23 ± 3 °C temperature, 50–70% humidity.

## Cholesterol and biochemical analysis of serum samples

Blood was collected from the eyeballs of mice. After standing at room temperature for 1 h, the whole blood was centrifuged at 2000 × $g$ for 20 min. The serum was then poured into new tubes for further analysis. The total cholesterol, albumin (ALB), and alanine aminotransferase (ALT) levels in serum were measured with a Catalyst One chemical analyzer (IDEXX, USA) following the manufacturer's protocol[44].

## Extraction and analysis of the RNA and protein indices of liver tissue

TRIzol (TIANGEN Biotech, China) was used to extract RNA from liver tissue, and a FastKing RT kit (with gDNase) (TIANGEN Biotech, China) was used to reverse transcribe the RNA. Thermo Fisher Scientific QuantStudio3 was used to perform quantitative PCR (qPCR) on liver tissue cDNA using SuperReal PreMix Plus (TIANGEN). AAV plasmid expression was measured, and *Pcsk9* mRNA knockout levels confirmed (Fig. S11). Liver tissues were fixed with 4% formalin overnight before being embedded in paraffin, sectioned, and stained with hematoxylin and eosin (H&E)[46]. The liver sections were prepared by Servicebio (https://www.servicebio.cn/).

## Western blotting

Western blotting was performed as previously described[56]. Briefly, whole-cell extracts were prepared with RIPA buffer and phenylmethanesulfonyl fluoride (PMSF, Beyotime Biotechnology) on ice for 30 min. The samples were vortexed briefly (10 min), and then subjected to western blot analyses with the anti-HA antibody (1:1000, H6908, Sigma), anti-GAPDH Polyclonal antibody (1:2000, 10494-1-AP, proteintech). The images were captured on the Tanon 5200 Imaging System (Tanon).

## Statistical analysis

All data are expressed as the mean ± s.d. of at least three replicates in all experiments. The editing efficiencies of different Cas enzymes in HEK293T cells were compared with GraphPad Prism software 8.0.1. Statistical significance is indicated by a probability value of 0.05 ($p < 0.05$). *$p < 0.05$, **$p < 0.01$, and ***$p < 0.001$.

## Reporting summary

Further information on research design is available in the Nature Portfolio Reporting Summary linked to this article.

## Data availability

The RNA-seq data and DNA sequences data generated in this study have been deposited in the SRA under BioProject PRJNA982412. The

Q-PCR data generated in this study are provided in the Source Data file. Source data are provided with this paper.

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

## Acknowledgements
This study was financially supported by the National Key Research and Development Program of China (2022YFA1105404), the National Natural Science Foundation of China (grant Nos. 32101226, 32170543 and 31970574), and the Young Elite Scientist Sponsorship Program by CAST (No. YESS20210189).

## Author contributions
F.Z. and T.Z. designed the project and with advice from L.L., T.S., and Z.L. carried out the simulations, and F.Z., T.Z., X.S., X.Z., L.C., and H.W. performed the experiments. F.Z., T.Z., H.W., and J.L. analyzed the data. F.Z. and T.Z. wrote the paper.

## Competing interests
The authors declare no competing interests.
