## [Peer Review File · Nature Communications]

Reviewers' Comments:

Reviewer #1:

Remarks to the Author:

This paper introduces an IDC strategy to engineer five mini variants of Cas13b and Cas13d. The authors show that these miniaturized proteins retained both full RNA-binding activity and full RNA-cleavage activity. Furthermore, the function of the mini variants was validated, confirming the mini variants' efficiency comparable to those of the WT protein in base-editing and in vivo experiments. This paper suggests a new strategy for structure design and a new use for AlphaFold.

However, it is not clear how the IDC strategy contributes to the advancement of protein miniaturization. It may be more appropriate to shift the focus of the paper toward the consistency of AlphaFold predicted miniaturization as compared to experimental based miniaturization.

Introduction:

It is not clearly stated whether the IDC strategy was developed by the authors (or based on another method). What are the other strategies already used in the protein design field? What is unique about this work? Also, each strategy I,D,C should be explained in a more intuitive manner to prepare readers to understand the complex concept. The current version reads more like a summary, not an introduction.

Results:

The values in Figure 1b, 1d should be shared in a table as it is almost impossible to read these bar charts (both averages and standard deviations).

Regarding the efficiency of mini-RfxCas13d-mediated knockdown in vivo, it will be helpful to compare the control and the mini variants with the efficiency of the previous non-modified CRISPR-Cas13d system.

Discussion:

Some discussion on the (potentially negative) impact of miniaturization on the protein function should be discussed. For example, how to optimize a protein that has gone through evolutionary optimization?

Materials and Methods:

There are no explanations on the steps for how the IDC strategy was used to optimize the

miniaturization of CRISPR-Cas13 systems. Each step should be explained with some theoretical/empirical reasoning for the choice of design, for any scientist to repeat or use the IDC strategy on a similar protein using these guidelines.

RNA sequence/structure aspects of DR were not included in the study, as it is important information for the CRISPR-Cas efficiency. For example, how was the 36-nt DR sequence selected?

Minor comments:

Figure resolution is generally poor - also fluorescent colors make it difficult to see structures.

Figures should be numbered chronologically.

Figure 6b,d: Please use another color than white as a minimum value.

Figure S16: Please use another color than white as a minimum conservation.

L67: "a thermal denaturation assay results"?

L177: can the "collateral effects" of these subtypes be explained a bit more in detail?

Reviewer #2:

Remarks to the Author:

Zhao et al. engineer five compact variants of Cas13 based on a combination of IDC strategy and AlphaFold2. Compared with the wild-type proteins, the size of these variants is significantly reduced, which makes it easier to deliver to cells using AAV. Additionally, the authors generate RNA base editor by fusing ADAR2 catalytic domain to the mini-RfxCas13d and demonstrate that the RNA base editor introduce point mutations in a programmable manner.

Overall, this study is very interesting. In particular, they reduced the size of Cas protein while retaining its nuclease activity. However, there are some aspects of this work that should be strengthened to enable the community to properly use and understand the new variant.

Major Comments:

1. The collateral activity of Cas13 variants is a major concern for the applications. Many studies reported that Cas13 can cause serious cytotoxicity. The authors should perform an unbiased specificity assay to compare the collateral activity of mini-RfxCas13d and the hfCas13d (across at least 2-3 target sites).
2. Furthermore, transcriptome-wide off-target assessment of the RNA base editor should also be performed (across at least one target sites).

3. Other functional features of mini-RfxCas13d should be compared with the wild-type RfxCas13 and hfCas13, such as best spacer length and mismatch tolerance.

4. Constructs and primary data. All primary data should be included in a supplementary table and made available at the time of publication. Furthermore, plasmids used in the study should be deposited with Addgene, to enable use of these new enzymes (and replication of data).

Minor Comments:

5. All the figures look ugly. Some improvements of the quality will be beneficial to readers. All data points should be clearly displayed on the histogram.

6. Some terms need to be unified and standardized, such as crRNA vs sgRNA, crPCSK9 vs sgPcsk9 and Cas13 vs CAS13. These will confuse readers.

7. Engineer the smaller Cas13s such as Cas13Xs or Cas13bts will enhance the innovation of this study.

Reviewer #3:

Type VI CRISPR system (Cas13 family) has become a model and powerful platform for RNA manipulation including detection, imaging, transcription knocking down and base editing. The diverse Cas13 family contains at least four known subtypes, including Cas13a, Cas13b, Cas13c, and Cas13d. The diversity be shedded in a large size range from 400 to 1600 amino acids. However, therapeutic delivery remains challenging because of the packaging capacity of adeno-associated virus (AAV), the most widely used viral vehicle for gene delivery. Thus, the compact Cas13 would benefit a lot for future therapeutic invention.

In this work, Feiyu Zhao and Tao Zhang et al., proposed a new strategy (**I**nteraction, **D**ynamics and **C**onservation, **IDC**) to generate a series of compact Cas13 variants and combined the deletion pattern into a smallest one called miniCas13 which showed only ~70% size of wildtype version. Then they applied this pipeline into five other homologs of Cas13 to produce mini Rfx-, mini Esi-, mini Rsp-, mini Pbu- and mini Psp-Cas13. They validated that the RNA knocking down activity from mini versions of Cas13 were comparable with full length Cas13(s) in human HEK293T cells. What's more, the authors developed a mini-RfxCas13d protein fusing with ADAR2 and APOBEC3A for RNA base editing which showed a robust base editing efficacy in cell and mice level.

Cas13 has already trimmed to ~ 415-750 amino acids without significant loss of activity (*Nature Biotechnology* 2022; *Nature Methods* 2022), however different Cas13 homologs display different characterizations, for example PFS sequence, stability, and precision etc. So this work exhibits some potential impact in certain extent. And the IDC strategy would be a possible example for protein optimization and minimization for some large proteins in the future. However, this reviewer is concerned that the concept, experiment design and conclusions are for most part incremental, and the insights do not significantly convince us, hence the authors should address these issues.

Major points:

1. The concept of IDC was not clear. Or it's hard to understand how to apply it. I didn't get any exact protocol or detail of how to do that in Main text and in Methods part. The authors just listed **I**nteraction, **D**ynamics and **C**onservation as three paragraphs, however, this is asymmetry with the article topic. The authors should give a figure panel in Figure 1 or Figure S1 to depict what's the pipeline or workflow for the new strategy called IDC. For example, what's the input, what's the output for each step. And so it is in Method part.

2. *In vivo* assays were filled in this work to validate the activity of smaller variants of Cas13. The data look elaborate, however, this is not enough to convince us for a new protein with so much trimming even in many different locations in one peptide chain. The *in vitro* biochemistry assays are required to convince us. The authors need to: A) purify at least one pair of wild type-Cas13 and mini-Cas13 side by side to verified the similar solubility and stability; B) do crRNA cleavage assay to validate the comparable crRNA processing ability with wt-Cas13; C) target RNA cleavage assay to verify its comparable RNA cleavage with wt-Cas13.
3. At least in one pair of RNA knock down assay, a western blot assay is necessary to confirm the protein level is similar or optimizable for wt- and mini-Cas13 in cell. The transfected amount of plasmids would produce the bias of protein expression level which is related with the RNA knockdown efficacy directly.
4. Given in some gene targeted locus, the mini-Cas13 even better than wt-Cas13, for example, Figure 1b $\Delta 4$ in LMNA, Figure 1d almost all, Figure 3c PPARG in et al., how to explain this correlation?
5. In line 284, the authors claim “the miniaturized Cas13 enzyme variants showed little effect on target cleavage activity and collateral cleavage activity”, however, in this work, there is not any data to indicate the difference of collateral activity between wt- and mini-Cas13. The authors claim collateral activity via off targeting effect results in mutation site (Figure 6c and 6d), that’s not the same concept. The authors should do collateral activity in *in vitro* or RNA-chip assay to assess all RNA level from edited cells, thus the author could give this conclusion.

Minor points:

1. Line 18, the terms for short should put into the brackets not before bracket for the first time. The author could list as Interaction-Dynamics-Conservation (IDC). The same to line 120.
2. Line 32, “The CRISPR–Cas system is a revolutionary gene-editing technology that shows precise targeting, high efficiency and programmable DNA- or RNA-targeting properties.”, this is an ill-formed sentence. CRISPR-Cas system is an adaptive immunity system, not a

technology, which are two different concepts. It can be described as “The CRISPR–Cas system has been developed/applied/adopted to be a revolutionary gene-editing technology that shows precise targeting, high efficiency and programmable DNA- or RNA-targeting properties”

3. Line 39-40, the cited cases should be much broadening if the author want to advocate small Cas enzyme. For example, CasMINI (Xu et al., Mol Cell 2021); Cas12f (Bigelyte et al., Nature Communications 2021); Compact Cas9 (Goltsman et al, Nature Communications); Tiny Cas9 (Altae-Tran et al., Science 2021; Schuler et al., Sciene 2022); Cas X/Y/ Φ (Pausch et., Science 2020; Liu et al., Nature 2019; Berstain et al., Nature 2016)...
4. Line 43, should be “amino acids”, not “AA”. Could be amino acids (AA), then AA can be used directly.
5. Line 68-70, wrong citation. I didn’t see “thermal denaturation assay” in the citation 28 and 29. And citation 28 is not an article for PbuCas13, it BzCas13. So this is a wrong citation.
6. The figure numbering is pretty confused at some parts. For example, at line 125-135, the depicting order is Figure b→e→f→d→a; Line 111, Figure S1→S2→S16. This is not illegal, however, this is really confused and hard for readers. Appreciate a lot if it could be changed as a normal ordering.

Response to reviewers:

Thank you very much for your comments concerning our manuscript entitled "A novel strategy for Cas13 miniaturization based on the structure". Those comments are all valuable and very helpful for revising and improving our paper, as well as the important guiding significance to our researches. We have studied comments carefully and made correction which we hope meet with approval. Revised portion are marked in blue in the "revised highlighted" copy. The main corrections in the paper and the responds to the comments are as flowing.

To Reviewer #1:

This paper introduces an IDC strategy to engineer five mini variants of Cas13b and Cas13d. The authors show that these miniaturized proteins retained both full RNA-binding activity and full RNA-cleavage activity. Furthermore, the function of the mini variants was validated, confirming the mini variants' efficiency comparable to those of the WT protein in base-editing and in vivo experiments. This paper suggests a new strategy for structure design and a new use for AlphaFold.

However, it is not clear how the IDC strategy contributes to the advancement of protein miniaturization. It may be more appropriate to shift the focus of the paper toward the consistency of AlphaFold predicted miniaturization as compared to experimental based miniaturization.

1. * Introduction:

It is not clearly stated whether the IDC strategy was developed by the authors (or based on another method). What are the other strategies already used in the protein design field? What is unique about this work? Also, each strategy I, D,C should be explained in a more intuitive manner to prepare readers to understand the complex concept. The current version reads more like a summary, not an introduction.

Response: Thank you for your good suggestion. IDC strategy is developed by the authors. We have added the description in line 77-108 of the revised manuscript.

There are no dedicated strategies for protein downsizing in the field of protein design currently. Previous strategies:

1. Deleting the entire domain^{1,2}

2. Generate a protein deletion library at random

3. Deleting surface portions with low conservation in protein³

The distinctive feature of this study lies in its ability to successfully miniaturize a cas protein for use in the future while maintaining complete protein activity with little loss of effectiveness. The picture had added in Figure S1 and described in line 77-108 and line 333-357 of the revised manuscript, and it provides a more intuitive explanation of each of the strategies I, D, and C.

2. *Results:

The values in Figure 1b, 1d should be shared in a table as it is almost impossible to read these bar charts (both averages and standard deviations).

Response: Thank you for this suggestion. The table was added in Supplementary table 6 of the revised manuscript.

Regarding the efficiency of mini-RfxCas13d-mediated knockdown in vivo, it will be helpful to compare the control and the mini variants with the efficiency of the previous non-modified CRISPR-Cas13d system.

Response: Thank you for this suggestion. The efficiency of mini-RfxCas13d-

mediated knockdown compare to RfxCas13d *in vivo* was shown in Figure 2b and 2c, and described in line 147-149 and line 151-152 of the revised manuscript.

3.*Discussion:

Some discussion on the (potentially negative) impact of miniaturization on the protein function should be discussed. For example, how to optimize a protein that has gone through evolutionary optimization?

Response: Thank you for your insightful and constructive comments and advice. The discussion on the impact of miniaturization on the protein function was added in line 322-324 of the revised manuscript.

4.*Materials and Methods:

There are no explanations on the steps for how the IDC strategy was used to optimize the miniaturization of CRISPR-Cas13 systems. Each step should be explained with some theoretical/empirical reasoning for the choice of design, for any scientist to repeat or use the IDC strategy on a similar protein using these guidelines.

Response: Thank you for the helpful suggestions. Detailed explanations on the steps for how the IDC strategy works were added in line 333-357 and Figure S1 of the revised manuscript. It is divided into four steps; each step is explained in detail and RfxCas13d is used as an example.

RNA sequence/structure aspects of DR were not included in the study, as it is important information for the CRISPR-Cas efficiency. For example, how was the 36-nt DR sequence selected?

Response: Thank you for pointing out this question. The description of the DR sequence selected has been added in line 145-147 of the revised manuscript according to the previous studies⁴⁻⁶. The specific sequence has been added in the supplementary table 3 of the revised manuscript.

5.*Minor comments:

Figure resolution is generally poor - also fluorescent colors make it difficult to see structures.

Response: Thank you for the suggestions. The figures resolution had been improved and fluorescent colors were changed in the revised manuscript.

Figures should be numbered chronologically.

Response: Thank you for the good suggestions. The figures have been

numbered chronologically in the revised manuscript (Figure 1 and Figure 4).

Figure 6b,d: Please use another color than white as a minimum value.

Response: Thank you very much for your good suggestion. The color of the minimum value has changed into light gray in Figure 6b and 6d of the revised manuscript.

Figure S16: Please use another color than white as a minimum conservation.

Response: Thank you very much for your good suggestion. The color of the minimum conservation has changed to turquoise in the revised manuscript.

L67: “a thermal denaturation assay results”?

Response: Thank you for pointing out this grammatical fault. The description has changed in line 70-74 of the revised manuscript.

L177: can the “collateral effects” of these subtypes be explained a bit more in detail?

Response: Thank you very much for your good suggestion. Cas13a-d and Cas13X have different levels of collateral effects. The collateral activities of Cas13 proteins refer to slicing non-target ssRNA after being activated. In view of the characteristic of Cas13 enzymes, many detection platforms were established to detect the RNA viruses, for example, SHERLOCK⁷ and CARVER⁸. However, the collateral effects also plagued the further application of gene therapy and drug development, etc.

1) Previous study has reported that PspCas13b has significantly improved specificity in *Drosophila* cells compared to RfxCas13d⁹. The eGFP and mCherry guide RNAs notably detected equally strong depletion (60–75%) of the non-target reporter mRNA as well as the mRNA encoding RfxCas13d itself. In stark contrast to RfxCas13d, the eGFP and mCherry guide RNAs tested with catalytically active PspCas13b resulted in no significant change in the expression of the non-target reporter mRNA or the mRNA encoding the Cas13 effector itself.

2) The collateral effects of Cas13X were lower than Cas13d, as the survival rate of mice with Cas13X was higher relative to Cas13d in previously study¹⁰.

To Reviewer #2:

Zhao et al. engineer five compact variants of Cas13 based on a combination of IDC strategy and AlphaFold2. Compared with the wild-type proteins, the size of

these variants is significantly reduced, which makes it easier to deliver to cells using AAV. Additionally, the authors generate RNA base editor by fusing ADAR2 catalytic domain to the mini-RfxCas13d and demonstrate that the RNA base editor introduce point mutations in a programmable manner.

Overall, this study is very interesting. In particular, they reduced the size of Cas protein while retaining its nuclease activity. However, there are some aspects of this work that should be strengthened to enable the community to properly use and understand the new variant.

Major Comments:

1. The collateral activity of Cas13 variants is a major concern for the applications. Many studies reported that Cas13 can cause serious cytotoxicity. The authors should perform an unbiased specificity assay to compare the collateral activity of mini-RfxCas13d and the hfCas13d (across at least 2-3 target sites).

Response: Thank you very much for your valuable suggestion. The results of the collateral activity of mini-RfxCas13d, and the hfRfxCas13d were presented and added in Figure.S6b-e, line 179-181 of the revised manuscript.

2. Furthermore, transcriptome-wide off-target assessment of the RNA base editor should also be performed (across at least one target sites).

Response: Thank you very much for your meaningful comments. The result of the transcriptome-wide off-target assessment of the RNA base editor was presented and added in Figure.S14, line 238-240 of the revised manuscript, according to your good suggestion.

3. Other functional features of mini-RfxCas13d should be compared with the wild-type RfxCas13 and hfCas13, such as best spacer length and mismatch tolerance.

Response: Thank you for your thoughtful suggestion. The result of spacer length and mismatch tolerance was presented and added in Figure.S5, and described in line 168-176 of the revised manuscript.

4. Constructs and primary data. All primary data should be included in a supplementary table and made available at the time of publication. Furthermore, plasmids used in the study should be deposited with Addgene, to enable use of these new enzymes (and replication of data).

Response: Thank you for your suggestion. All primary data were presented in supplementary Table 6.

Plasmids used in the study would be deposited to Addgene.

Minor Comments:

5. All the figures look ugly. Some improvements of the quality will be beneficial to readers. All data points should be clearly displayed on the histogram.

Response: Thank you for your suggestion. The quality of the figures had improved in the revised manuscript. The figures' color had changed in the revised manuscript.

6. Some terms need to be unified and standardized, such as crRNA vs sgRNA, crPCSK9 vs sgPcsk9 and Cas13 vs CAS13. These will confuse readers.

Response: I apologize for these mistakes in terms. "SgRNA" had changed into "crRNA" in line 381, and "sgPcsk9" had changed into "crPCSK9" line 703, line 704, and line 705 of the revised manuscript.

7. Engineer the smaller Cas13s such as Cas13Xs or Cas13bts will enhance the innovation of this study.

Response: Thank you for your insightful and constructive comments and advice. We engineer mini-Cas13bt3 by the IDC strategy, and the Q-PCR results show mini-Cas13bt3 has an efficiency loss relative to Cas13bt3 (Figure 1).

Figure 1 Comparison of the Cas13bt3 and mini-Cas13bt3 knockdown efficiency of endogenous transcripts.

We found that the efficiency of mini-Cas13bt3 was decreased compared with Cas13bt3. Given that the result, we guessed that it was resulted as the size of Cas13bt3 was small enough that the structure of Cas13bt3 is extremely compact (Figure 2). Thus, when we tried to miniaturize the Cas13bt3 by IDC strategy, mini-Cas13bt3 inevitably lost a part of knockdown efficiency.

Figure 2 The overall structure of Cas13bt3 (PDB:7VTI) and mini-Cas13bt3.

To Reviewer #3:

Type VI CRISPR system (Cas13 family) has become a model and powerful platform for RNA manipulation including detection, imaging, transcription knocking down and base editing. The diverse Cas13 family contains at least four known subtypes, including Cas13a, Cas13b, Cas13c, and Cas13d. The diversity be shedded in a large size range from 400 to 1600 amino acids.

However, therapeutic delivery remains challenging because of the packaging capacity of adeno-associated virus (AAV), the most widely used viral vehicle for gene delivery. Thus, the compact Cas13 would benefit a lot for future therapeutic invention.

In this work, Feiyu Zhao and Tao Zhang et al., proposed a new strategy (Interaction, Dynamics and Conservation, IDC) to generate a series of compact Cas13 variants and combined the deletion pattern into a smallest one called miniCas13 which showed only ~70% size of wildtype version. Then they applied this pipeline into five other homologs of Cas13 to produce mini Rfx-, mini Esi-, mini Rsp-, mini Pbu- and mini Psp-Cas13. They validated that the RNA knocking down activity from mini versions of Cas13 were comparable with full length Cas13(s) in human HEK293T cells. What's more, the authors developed a mini-RfxCas13d protein fusing with ADAR2 and APOBEC3A for RNA base editing which showed a robust base editing efficacy in cell and mice level.

Cas13 has already trimmed to ~ 415-750 amino acids without significant loss of activity (Nature Biotechnology 2022; Nature Methods 2022), however different Cas13 homologs display different characterizations, for example PFS

sequence, stability, and precision etc. So this work exhibits some potential impact in certain extent. And the IDC strategy would be a possible example for protein optimization and minimization for some large proteins in the future. However, this reviewer is concerned that the concept, experiment design and conclusions are for most part incremental, and the insights do not significantly convince us, hence the authors should address these issues.

Major points:

1. The concept of IDC was not clear. Or it's hard to understand how to apply it. I didn't get any exact protocol or detail of how to do that in Main text and in Methods part. The authors just listed Interaction, Dynamics and Conservation as three paragraphs, however, this is asymmetry with the article topic. The authors should give a figure panel in Figure 1 or Figure S1 to depict what's the pipeline or workflow for the new strategy called IDC. For example, what's the input, what's the output for each step. And so it is in Method part.

Response: Thank you very much for your good suggestion. The depict what's the pipeline or workflow for the new strategy called IDC has been added in Figure S1 and described in line 333-357 of the revised manuscript.

2. In vivo assays were filled in this work to validate the activity of smaller variants of Cas13. The data look elaborate, however, this is not enough to convince us for a new protein with so much trimming even in many different locations in one peptide chain. The in vitro biochemistry assays are required to convince us. The authors need to: A) purify at least one pair of wild type-Cas13 and mini-Cas13 side by side to verified the similar solubility and stability; B) do crRNA cleavage assay to validate the comparable crRNA processing ability with wt-Cas13; C) target RNA cleavage assay to verify its comparable RNA cleavage with wt-Cas13.

Response: Thank you for such a good suggestion.

1) The result of purifying RfxCas13d, mini-RfxCas13d, PspCas13b, and mini-PspCas13b proteins were added in Figure S3 and Figure S4a-d, and described in line 157-161, and line 209-211 of the revised manuscript.

2) The crRNA cleavage assay was added in Figure S4e and S4f, and described in line 157-167, and line 209-214 of the revised manuscript.

3) The target RNA cleavage assay was added in Figure S4g and S4h, and described in line 157-167, and line 209-214 of the revised manuscript.

3. At least in one pair of RNA knock down assay, a western blot assay is necessary to confirm the protein level is similar or optimizable for wt- and mini-Cas13 in cell. The transfected amount of plasmids would produce the bias of protein expression level which is related with the RNA knockdown efficacy directly.

Response: Thank you for your thoughtful suggestion. The western blot assays have been added in Figure S2 and described in line 152-153 and line 215-217 of the revised manuscript.

4. Given in some gene targeted locus, the mini-Cas13 even better than wt-Cas13, for example, Figure 1b $\Delta 4$ in LMNA, Figure 1d almost all, Figure 3c PPARG in et al., how to explain this correlation?

Response: Thank you for your good question. We speculate that site specificity has an influence on knockdown efficiency. Also, different spacer sequence of crRNA at the same site will have different knockdown efficiencies¹⁰.

From reference 49 “High-fidelity Cas13 variants for targeted RNA degradation with minimal collateral effect”, hFRfxCas13d has higher knockdown efficiency than RfxCas13d with g3 and g4 evidently, and slight lower knockdown efficiency with g1, g6, and g7 in CA2.

In this study, the knockdown efficacy of $\Delta 4$ was equivalent at the NRAS, EGFR, and STAT6 sites, but higher at LMNA sites than RfxCas13d. Mini-EsCas13d performed better than EsCas13d at the PPARG site but performed worse than EsCas13d at the MATAL1 and NRAS sites. We suspect that various spacer sequences may impact protein efficiency.

5. In line 284, the authors claim “the miniaturized Cas13 enzyme variants showed little effect on target cleavage activity and collateral cleavage activity”, however, in this work, there is not any data to indicate the difference of collateral activity between wt- and mini-Cas13. The authors claim collateral activity via off targeting effect results in mutation site (Figure 6c and 6d), that’s not the same concept. The authors should do collateral activity in in vitro or RNA-chip assay to assess all RNA level from edited cells, thus the author could give this conclusion.

Response: Thank you very much for your good suggestion. The collateral activity *in vitro* has been added in Figure S6 and S14 and described in line 176-181 and line 238-240 of the revised manuscript.

Minor points:

1. Line 18, the terms for short should put into the brackets not before bracket for the first time. The author could list as Interaction-Dynamics-Conservation (IDC). The same to line 120.

Response: Thank you very much for your good suggestion. We have changed to “Interaction-Dynamics-Conservation (IDC)” in line 20 of the revised manuscript.

2. Line 32, “The CRISPR–Cas system is a revolutionary gene-editing technology that shows precise targeting, high efficiency and programmable DNA- or RNA-targeting properties.”, this is an ill-formed sentence. CRISPR-Cas system is an adaptive immunity system, not a technology, which are two different concepts. It can be described as “The CRISPR–Cas system has been developed/applied/adopted to be a revolutionary gene-editing technology that shows precise targeting, high efficiency and programmable DNA- or RNA-targeting properties”

Response: Thank you very much for your point question. We have changed line 34 of the revised manuscript.

3. Line 39-40, the cited cases should be much broadening if the author want to advocate small Cas enzyme. For example, CasMINI (Xu et al., Mol Cell 2021); Cas12f (Bigelyte et al., Nature Communications 2021); Compact Cas9 (Goltsman et al, Nature Communications); Tiny Cas9 (Altae-Tran et al., Science 2021; Schuler et al., Science 2022); Cas X/Y/ Φ (Pausch et., Science 2020; Liu et al., Nature 2019; Berstain et al., Nature 2016)...

Response: Thank you very much for pointing out this question. We have changed line 41-42 of the revised manuscript.

4. Line 43, should be “amino acids”, not “AA”. Could be amino acids (AA), then AA can be used directly.

Response: Thank you very much for pointing out this question. We have changed line 45 of the revised manuscript.

5. Line 68-70, wrong citation. I didn’t see “thermal denaturation assay” in the citation 28 and 29. And citation 28 is not an article for PbuCas13, it BzCas13. So this is a wrong citation.

Response: Thank you very much for pointing out this question. The description has changed in line 69-74 in the revised manuscript.

6. The figure numbering is pretty confused at some parts. For example, at line 125-135, the depicting order is Figure b□e□f□d□a; Line 111, Figure S1□S2□S16. This is not illegal, however, this is really confused and hard for readers. Appreciate a lot if it could be changed as a normal ordering.

Response: Thank you very much for your good suggestion. The figure numbering has changed chronologically in the revised manuscript.

Thank you very much for your email and kindly suggestion. We are glad to receive further information about the manuscript. We would try our best to edit the manuscript to fit for the requirement of *Nature Communications*.

My contact information is: E-mail: lai_liangxue@gibh.ac.cn; Tel: (86) 431-87836175; Fax: (86) 431-87980131.

Sincerely Yours,

Liangxue Lai

References

1. Ma, D., Peng, S., Huang, W., Cai, Z. & Xie, Z. Rational Design of Mini-Cas9 for Transcriptional Activation. *ACS Synth Biol* **7**, 978–985 (2018).
2. Villiger, L. et al. Replacing the SpCas9 HNH domain by deaminases generates compact base editors with an alternative targeting scope. *Mol Ther Nucleic Acids* **26**, 502–510 (2021).
3. Shams, A. et al. Comprehensive deletion landscape of CRISPR-Cas9 identifies minimal RNA-guided DNA-binding modules. *Nat Commun* **12**, 5664 (2021).

4. Konermann, S. et al. Transcriptome Engineering with RNA-Targeting Type VI-D CRISPR Effectors. *Cell* **173**, 665-676 e14 (2018).
5. Smargon, A.A. et al. Cas13b Is a Type VI-B CRISPR-Associated RNA-Guided RNase Differentially Regulated by Accessory Proteins Csx27 and Csx28. *Molecular Cell* **65**, 618-+ (2017).
6. Yan, W.X. et al. Cas13d Is a Compact RNA-Targeting Type VI CRISPR Effector Positively Modulated by a WYL-Domain-Containing Accessory Protein. *Molecular Cell* **70**, 327-+ (2018).
7. Gootenberg, J.S. et al. Multiplexed and portable nucleic acid detection platform with Cas13, Cas12a, and Csm6. *Science* **360**, 439-+ (2018).
8. Freije, C.A. et al. Programmable Inhibition and Detection of RNA Viruses Using Cas13. *Mol Cell* **76**, 826-837 e11 (2019).
9. Ai, Y.X., Liang, D.M. & Wilusz, J.E. CRISPR/Cas13 effectors have differing extents of off-target effects that limit their utility in eukaryotic cells. *Nucleic Acids Research* **50**(2022).
10. Tong, H.W. et al. High-fidelity Cas13 variants for targeted RNA degradation with minimal collateral effects. *Nature Biotechnology* (2022).

Reviewers' Comments:

Reviewer #2:

Remarks to the Author:

Remarks to the Authors:

Zhao et al. present a revised manuscript that addresses many of the original comments, producing more data to support characterization of the miniCas13 generated by the IDC strategy. However, this reviewer has still some comments/suggestions in the light of new data presented.

Major concerns

1. From the new data in Fig. S6, the mini-RfxCas13d induced much more differentially expressed genes than that of hfRfxCas13d even when knockdown two separate genes expressed at low levels, indicating high collateral activity of mini-RfxCas13d. It is good to see that mini-RfxCas13d, in which the N3V7 fragment is deleted, induced less differentially expressed genes than that of RfxCas13d. The collateral activity is a major concern for biosafety and could hinder the future in vivo applications of mini-RfxCas13d. The authors could test to introduce the mutations in N2V7 or N2V8 variants into mini-RfxCas13d to generate miniaturized variants with high specificity. A miniaturized Cas13 with high specificity would be more useful to the field.

2. The authors highlighted in the title and abstract that they established IDC as a novel strategy for protein miniaturization. Although the authors added the description of Interaction, Dynamics and Conservation, respectively, it's still hard for readers to understand the concept of IDC strategy and how to apply it. More details of this key methodology should be provided. The authors should state the quantitative and/or qualitative assessment of the Interaction, Dynamics and Conservation, that is why they chose to delete each of the fragment (e.g., $\Delta 1 - \Delta 8$ in Fig. 1). More clear statement about this would help to rationally explain the failure of Cas13bt3 miniaturization, and also help others to implement the strategy in new system.

3. As far as I know, the structure of RfxCas13d has not been experimentally resolved, and the applications of the IDC strategy were based on structure predicted by AlphaFold2. So, the Interaction and Dynamics are not applicative for RfxCas13d. For the Conservation, the authors did not compare RfxCas13d with any other Cas13d orthologs. EsCas13d was shown as an example in Fig.S1 IDC strategy workflow. Thus, miniaturization of RfxCas13d may be not suitable for the illustration of the IDC strategy. Apart from multiple sequence alignment of Cas13 proteins in Fig. S18-S19, structure comparison of RfxCas13d and other Cas13d (align two or more Cas13d structures, overlaid structures with EsCas13d at least) should be presented alongside the explanatory description of RfxCas13d miniaturization, and this point is also applicable for that of PspCas13b.

Minor concerns:

1. Line 42, IscB, IsrB and TnpB were collectively referred as OMEGA, so it is improper to list OMEGA together with IscB right here.
2. Line 45, AAs could be directly used for the second amino acids (AAs).
3. Line 75-76 and Line 109-110 are repetitive.
4. Structure of RfxCas13d predicted by AlphaFold2 should be stated in the main text of Line 121-141.
5. Fig. 2b,c, Please clarify "WT". What is the control?
6. Fig. 2c, the name of the statistical test should be stated, note that when analyze data with 3 or more conditions, ANOVA analysis should be conducted. Please change it in all panels (main and supp figures) that it has not been properly done.
7. Fig.4c, PspCas13d should be PspCas13b?
8. Fig. 6e, mini-Vx should be mini-RfxCas13d?
9. Fig. 6e-j, PCSK9 in the figure panel should be Pcsk9, in line with that in the main text and legend.
10. Line 423, VX should be Vx?
11. There seems to be some incorrectly displayed symbol ("+") for page numbers in the reference list.
12. Fig. S6, NF2 and B4GALNT1 should be labelled in the corresponding volcano plot.
13. Line 348, Figure17 should be Fig. S17?
14. Fig. S18-S19 should be stated in the main text or section of Materials and Methods.

Reviewer #3:

Remarks to the Author:

Comments from Reviewer #1

Thanks for the authors' effort in making this article more consistent and solid.

1. In the introduction part, the authors revised the paragraph to introduce the IDC strategy. The current version reads more like an introduction than the original version.

2. In the result part, the authors resolved my two concerns well. First, in this version, an additional table was added in Supplementary Table 6 for the original raw data. Secondly, they added two control assays in Fig 2b and 2c, which makes the efficacy comparable between mini variants and previous non-modified CRISPR-Cas13d system.

3. In the discussion part, they also supplement the potentially negative impact of miniaturization on protein function. This would be an objective discussion on protein miniaturization.

4. In the methods part, they added the method description about using the IDC strategy to optimize the miniaturization of CRISPR-Cas13 systems. This helps give scientists worldwide to handle this new strategy for protein miniaturization. Meanwhile, they added DR's RNA sequence/structure aspects, making this work clearer.

5. In the minor's comments, the authors emended them well to make this version much more fluent. Especially more details of collateral activity were supplemented in this revised version.

Overall, in this revised version, by incorporating additional experiments and thoroughly addressing all concerns raised by the reviewers, the authors have significantly strengthened their research's overall impact and validity. This reviewer thinks the revisions are satisfactory and sincere.

At last, here this reviewer has one more insist from the last round of review: It may be more appropriate to use the title "A novel strategy for Cas13 miniaturization based on AlphaFold". It may make the article more distinct as compared to experimental-based miniaturization.

Comments from Reviewer #3

This reviewer truly impressed with the extensive improvements the authors have implemented in response to the feedback provided during the initial review. The authors' diligent efforts have resulted in a significantly enhanced paper that further strengthens its contribution to the field. They carefully considered each suggestion and incorporated them thoughtfully into the revised manuscript. One of the most notable improvements can be observed in the supplementals. The revised version now presents a clear and soluble protein purification of miniCas13(s) which display active activity of crRNA processing. This convince this reviewer to believe the editing efficacy of the minimized Cas13(s). The methodology section of IDC has undergone significant improvements, showcasing the commitment to rigor and transparency.

One minor suggestion, I agree with other two reviewers, the figure resolution and quality look not good. The color blending looks not soft. So it can be enhanced more.

Response to reviewers:

Thank you very much for your comments concerning our manuscript entitled “A novel strategy for Cas13 miniaturization based on the structure”. Those comments are all valuable and very helpful for revising and improving our paper, as well as the important guiding significance to our researches. We have studied comments carefully and made correction which we hope meet with approval. Revised portion are marked in blue in the "revised highlighted" copy. The main corrections in the paper and the responds to the comments are as following.

To Reviewer #2:

Zhao et al. present a revised manuscript that addresses many of the original comments, producing more data to support characterization of the miniCas13 generated by the IDC strategy. However, this reviewer has still some comments/suggestions in the light of new data presented.

Major concerns:

1. From the new data in Fig. S6, the mini-RfxCas13d induced much more differentially expressed genes than that of hfRfxCas13d even when knockdown two separate genes expressed at low levels, indicating high collateral activity of mini-RfxCas13d. It is good to see that mini-RfxCas13d, in which the N3V7 fragment is deleted, induced less differentially expressed genes than that of RfxCas13d. The collateral activity is a major concern for biosafety and could hinder the future in vivo applications of mini-RfxCas13d. The authors could test to introduce the mutations in N2V7 or N2V8 variants into mini-RfxCas13d to generate miniaturized variants with high specificity. A miniaturized Cas13 with high specificity would be more useful to the field.

Response: Thank you for your insightful and constructive comments and advice. We introduced the mutation in N2V8 variants into mini-RfxCas13d to generate mini-hfRfxCas13d. The transcriptome-wide off-target results were added in Figure S6d, and described in line 195-198,339-342 of the revised manuscript. To our surprise, mini-hfRfxCas13d and mini-RfxCas13d both had comparable numbers of differentially expressed genes in the transcriptome. We reasoned that the deletion of N3V7 or other fragments may have an impact on the function of N2V8 variant and cause diminished performance in high fidelity relative to N2V8 variants.

2. The authors highlighted in the title and abstract that they established IDC as

a novel strategy for protein miniaturization. Although the authors added the description of Interaction, Dynamics and Conservation, respectively, it's still hard for readers to understand the concept of IDC strategy and how to apply it. More details of this key methodology should be provided. The authors should state the quantitative and/or qualitative assessment of the Interaction, Dynamics and Conservation, that is why they chose to delete each of the fragment (e.g., $\Delta 1 - \Delta 8$ in Fig. 1). More clear statement about this would help to rationally explain the failure of Cas13bt3 miniaturization, and also help others to implement the strategy in new system.

Response: Thank you for your thoughtful suggestion. The details of IDC strategy were added in line128-158 and five videos (corresponding to five domains of EsCas13d) were provided of the revised manuscript.

3. As far as I know, the structure of RfxCas13d has not been experimentally resolved, and the applications of the IDC strategy were based on structure predicted by AlphaFold2. So, the Interaction and Dynamics are not applicable for RfxCas13d. For the Conservation, the authors did not compare RfxCas13d with any other Cas13d orthologs. EsCas13d was shown as an example in Fig.S1 IDC strategy workflow. Thus, miniaturization of RfxCas13d may be not suitable for the illustration of the IDC strategy. Apart from multiple sequence alignment of Cas13 proteins in Fig. S18-S19, structure comparison of RfxCas13d and other Cas13d (align two or more Cas13d structures, overlaid structures with EsCas13d at least) should be presented alongside the explanatory description of RfxCas13d miniaturization, and this point is also applicable for that of PspCas13b.

Response: Thank you for your thoughtful suggestion. The description of the RfxCas13d and EsCas13d has been amended in line 125-130,132-138,139-143, and 144-151 of the revised manuscript.

The structural distinctions in the overlap are less obvious since they share the same domain color scheme (Figure 1);

Figure 1. Alignment of EsCas13d (PDB:6E9E) and RfxCas13d.

The structural differences are still not readily apparent, even after the transparency distinction has been made (Figure 2);

Figure 2. Alignment of EsCas13d (PDB:6E9E) and RfxCas13d (transparent 70%).

With variable color matching, the protein structural contrasts were not apparent (Figure 3);

Figure 2. Alignment of EsCas13d (PDB:6E9E, blue) and RfxCas13d (yellow). The protein domains use different color scheme, that is similar to the first version, which is more dazzling, so we compared domains separately.

Minor Comments:

1. Line 42, IscB, IsrB and TnpB were collectively referred as OMEGA, so it is improper to list OMEGA together with IscB right here.

Response: Thank you very much for your good suggestion. “IscB” has deleted in line 42 of the revised manuscript.

2. Line 45, AAs could be directly used for the second amino acids (AAs).

Response: Thank you for pointing out this grammatical fault. The second amino acids (AAs) had replaced by AAs in line 46 of the revised manuscript.

3. Line 75-76 and Line 109-110 are repetitive.

Response: Thank you for the good suggestions. The description in line 109-110 has changed in line 110-112 of the revised manuscript.

4. Structure of RfxCas13d predicted by AlphaFold2 should be stated in the main text of Line 121-141.

Response: Thank you for the good suggestions. The description of AlphaFold2 has added in line 128-130 of the revised manuscript.

5. Fig. 2b,c, Please clarify “WT”. What is the control?

Response: Thank you for pointing out this question. “WT” had changed into

“NT” in Fig 2b,c ;Fig3e,f ; Fig5e,f and FigS15a. The description of “NT” has been added in line 648,651-652,671,693 of the revised manuscript and line 99 of the revised supplementary information.

6. Fig. 2c, the name of the statistical test should be stated, note that when analyze data with 3 or more conditions, ANOVA analysis should be conducted. Please change it in all panels (main and supp figures) that it has not been properly done.

Response: Thank you for your thoughtful suggestion. The name of the statistical test has been added in line 649-650 (Fig 2b), 652-653 (Fig 2c), 671-673 (Fig 3e,f), 694-695 (Fig 5e,f), 701-702 (Fig 6b), 707-708 (Fig 6f),712-713 (Fig 6h-j) of the revised manuscript and line 45-46 (Fig S5), 99-101 (Fig S15a,b) of the revised supplementary information.

7. Fig.4c, PspCas13d should be PspCas13b?

Response: Thank you very much for pointing out this question. The name has been changed in Fig 4c of the revised manuscript.

8. Fig. 6e, mini-Vx should be mini-RfxCas13d?

Response: Thank you very much for pointing out this question. The name has been changed in Fig 6e of the revised manuscript.

9. Fig. 6e-j, PCSK9 in the figure panel should be Pcsk9, in line with that in the main text and legend.

Response: Thank you very much for pointing out this question. PCSK9 in the figure panel has been changed in Fig 6e-j of the revised manuscript and FigS15a of the revised supplementary information.

10. Line 423, VX should be Vx?

Response: Thank you very much for pointing out this question. VX has been changed in Line 441 of the revised manuscript.

11. There seems to be some incorrectly displayed symbol (“+”) for page numbers in the reference list.

Response: Thank you very much for pointing out this question. The reference list has changed in Line 498-619 of the revised manuscript.

12. Fig. S6, NF2 and B4GALNT1 should be labelled in the corresponding volcano plot.

Response: Thank you very much for your good suggestion. NF2 and B4GALNT1 has been labelled in Fig S6 of the revised manuscript.

13. Line 348, Figure17 should be Fig. S17?

Response: Thank you very much for pointing out this question. The name has been changed in line 366 of the revised manuscript.

14. Fig. S18-S19 should be stated in the main text or section of Materials and Methods.

Response: Thank you very much for your good suggestion. The description has been added in line 152-153 and line 367 of the revised manuscript.

To Reviewer #3:

Minor Comments:

1. It may be more appropriate to use the title "A novel strategy for Cas13 miniaturization based on AlphaFold". It may make the article more distinct as compared to experimental-based miniaturization.

Response: Thank you for your thoughtful suggestion. The title has been changed in line 1 of the revised manuscript.

2. One minor suggestion, I agree with other two reviewers, the figure resolution and quality look not good. The color blending looks not soft. So it can be enhanced more.

Response: Thank you for your suggestion. The resolution of the figures has been improved in Figure1-6 of the revised manuscript. The color blending of Figure 6 has been changed of the revised manuscript.

Reviewers' Comments:

Reviewer #2:

Remarks to the Author:

Thanks to the authors for responding to my concerns by performing more experiments and providing detailed explanations for RfxCas13d miniaturization using the IDC strategy. The manuscript is much improved. With some minor modifications, this work would be acceptable for publication.

Misspelling for (23) in Line 283?

"WT" should be "NT" in Fig. 3e,f; Fig. 5c,d.

Response to reviewers:

Thank you very much for your comments concerning our manuscript entitled “A novel strategy for Cas13 miniaturization based on the structure”. Those comments are all valuable and very helpful for revising and improving our paper, as well as the important guiding significance to our researches. We have studied comments carefully and made correction which we hope meet with approval. Revised portion are marked in blue in the "revised highlighted" copy. The main corrections in the paper and the responds to the comments are as flowing.

To Reviewer #2:

Thanks to the authors for responding to my concerns by performing more experiments and providing detailed explanations for RfxCas13d miniaturization using the IDC strategy. The manuscript is much improved. With some minor modifications, this work would be acceptable for publication.

Minor Comments:

1. Misspelling for (23) in Line 283?

Response: Thank you for pointing out this spelling fault. “(23)” had changed in line 278 of the revised manuscript.

2. “WT” should be “NT” in Fig. 3e,f; Fig. 5c,d.

Response: Thank you for pointing out this question. “WT” had changed into “NT” in Fig3e,f ; Fig5c,d.